# A robust method for collider bias correction in conditional genome-wide association studies

Osama Mahmoud [1,2✉], Frank Dudbridge [3], George Davey Smith [4,5], Marcus Munafo [4,6] & Kate Tilling [4,5]

Estimated genetic associations with prognosis, or conditional on a phenotype (e.g. disease incidence), may be affected by collider bias, whereby conditioning on the phenotype induces associations between causes of the phenotype and prognosis. We propose a method, 'Slope-Hunter', that uses model-based clustering to identify and utilise the class of variants only affecting the phenotype to estimate the adjustment factor, assuming this class explains more variation in the phenotype than any other variant classes. Simulation studies show that our approach eliminates the bias and outperforms alternatives even in the presence of genetic correlation. In a study of fasting blood insulin levels (FI) conditional on body mass index, we eliminate paradoxical associations of the underweight loci: *COBLLI*; *PPARG* with increased FI, and reveal an association for the locus rs1421085 (*FTO*). In an analysis of a case-only study for breast cancer mortality, a single region remains associated with more pronounced results.

[1] Department of Mathematical Sciences, University of Essex, Colchester, UK. [2] Department of Applied Statistics, Helwan University, Helwan, Egypt. [3] Department of Health Sciences, University of Leicester, Leicester, UK. [4] MRC Integrative Epidemiology Unit, University of Bristol, Bristol, UK. [5] Department of Population Health Sciences, Bristol Medical School, University of Bristol, Bristol, UK. [6] School of Psychological Science, University of Bristol, Bristol, UK. ✉email: o.mahmoud@essex.ac.uk

There is increasing interest in the use of genome-wide association studies (GWASs) conditioned on a phenotype, such as a GWAS of blood insulin conditional on body mass index (BMI)[1] so as to avoid only genetic variants associated with BMI appearing important. An example of such conditional analyses is GWAS of prognosis[2–5]. Studies of prognosis, of necessity, can be conducted only in those who have the disease, i.e. conditioning on the disease status[6]. Such an analysis is referred to as 'conditional analysis' throughout this manuscript. This leads to a type of selection bias—termed index event bias or collider bias—whereby uncorrelated causes of the disease appear correlated when carrying out a conditional analysis, or studying only cases[2,3,7,8]. This means that if there is unmeasured confounding between incidence and prognosis, then any cause of incidence will appear also to cause prognosis. Any cause of both incidence and prognosis will have a biased estimate of its effect on prognosis. The direction and size of the bias depend on the incidence mechanism—with no collider bias if all factors affect incidence independently (i.e. if the incidence model is additive on the log probability scale).

Figure 1a, b, c illustrate that a single nucleotide polymorphism (SNP), $G$, causing a trait $I$ becomes correlated with the confounder, $U$, of $I$ and the outcome, $P$, when conditioning on $I$. This induces an association between $G$ and $P$ via the path $G − U \rightarrow P$ leading to collider bias in the SNP-outcome association, if the confounding effects are not accounted for. If all common causes ($U$) of $I$ and $P$ were known and could be measured, the collider bias could then be removed, e.g. by using the inverse probability weighting (IPW) approach[9]. But, for IPW to be valid, the weighting model must be correctly specified, and must include all variables that are related to both incidence and to the variables in

the analysis model (e.g., the outcome and every genetic variant). However in most studies, these variables are not all known, and not all are measured. Collider bias only affects causes of the variable conditioned on. Thus, in case-only studies, collider bias only affects causes of disease. As illustrated in Fig. 1d, e, associations of SNPs that do not cause $I$ with the outcome conditional on $I$ would not suffer from the collider bias problem, due to the lack of association induced with $U$.

The implications of collider bias have been addressed in several GWAS and MR studies[2,3]. An example is the 'paradox of glucose-6-phosphate dehydrogenase (G6PD) deficiency' whereby among individuals selected according to their status of severe malarial anaemia (SMA), higher levels of G6PD deficiency appear to protect against cerebral malaria (CM)[10,11]. A possible explanation is that if an individual with SMA has a high level of G6PD deficiency, they may well have lower levels of other risk factors for SMA. If lower levels of those other factors tend to decrease the risk of CM, then the G6PD deficiency may appear to be protective against CM. In the notation of Fig. 1a, G6PD deficiency plays the role of the SNP $G_I$, whereas $I$ and $P$ represent SMA and CM, respectively. It has been suggested that this apparent protective effect is at least partially due to collider bias[6].

A method for adjusting GWAS of disease progression for collider bias has been proposed whereby estimated residuals from the regression of SNP-outcome associations on SNP-incidence associations give bias-adjusted associations with outcome[7]. This method assumed that the genetic effects on incidence and direct genetic effects on outcome are linearly uncorrelated. But this assumption may be incompatible with most genetic studies where shared pathways have been observed for many traits including psychiatric[12], metabolites[13] and phenotypes related to cumulative effects of long-term exposures[14].

When conditioning on a quantitative trait, the direct SNP-outcome associations could be obtained by using Mendelian randomisation (MR) to estimate the causal effect of $I$ on $P$ and then subtracting the $G \rightarrow I \rightarrow P$ path from the total $G \rightarrow P$ association[15]. This approach, implemented in the mtCOJO software, presumes a causal effect of I on P and the availability of unconditional $G \rightarrow P$ effects, but is not applicable to case-only studies of disease progression.

We propose an alternative method, referred to as 'Slope-Hunter', for adjustment of collider bias in GWAS of conditional analyses (including index event bias in progression studies) with potentially correlated direct genetic effects on incidence and outcome. This is achieved by first identifying the set of SNPs which only affect the incidence, and then using it to obtain an unbiased estimate of the correction factor that is then used to adjust for the bias for all genetic variants. We evaluate the Slope-Hunter method by comparing its type-1 error, power and bias with the naive (unadjusted) conditional analyses and previously proposed methods in an extensive simulation study with realistic parameters. We illustrate our method in a GWAS of fasting insulin levels conditional on BMI[1] and a GWAS of survival with breast cancer[5].

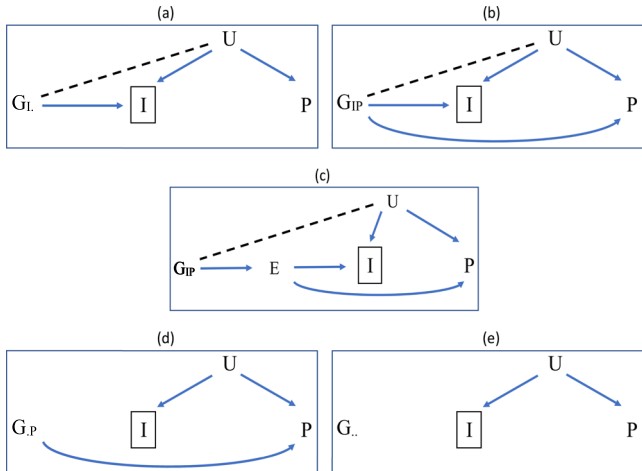

**Fig. 1 Directed acyclic graph for associations of different SNPs with an outcome P conditional on a trait I.** (**a**) association of a SNP ($G_I$) with an outcome P conditional on a trait I such that the SNP $G_I$ affects I with no direct effect on P; (**b**) association of a SNP ($G_{IP}$) with P conditional on I such that the SNP $G_{IP}$ has direct effects on both I and P; (**c**) as in (**b**) but the SNP $G_{IP}$ affects both I and P through a single exposure E; (**d**) association of a SNP ($G_P$) with P conditional on I such that the SNP $G_P$ affects P with no effect on I; (**e**) a SNP ($G_{..}$) with neither effects on I nor P. In all graphs, U is a composite variable including all common causes of I and P, involving common polygenic effect on I and P as well as non-genetic factors. Conditioning on I induces the association between $G_I$ and U in (**a**), as well as $G_{IP}$ and U in (**b**) and (**c**), shown by the dashed lines. This leads to biased association for each of $G_I$ and $G_{IP}$ with P via the path $G_I − U \rightarrow P$ in (**a**) and the path $G_{IP} − U \rightarrow P$ in (**b**) and (**c**). Since the SNPs $G_P$ and $G_{..}$ do not affect I, conditioning on I does not induce biased association between either SNP and P as no associations between $G_P$ or $G_{..}$ and U are produced.

## Results

**Simulations.** Simulation studies show that the Slope-Hunter method eliminates or minimises the collider bias, outperforming the alternative methods under a wide range of scenarios (Methods) when its assumptions were satisfied (Figs. 2, 3, 4).

When averaged over all variants, the standard unadjusted analysis as well as the adjusted analyses using Slope-Hunter and using the method of Dudbridge et al.[7] (DHO) generally give type-1 error rates that are close to the nominal level, 0.05 (Supplementary Tables 1–3). Since the majority of genetic

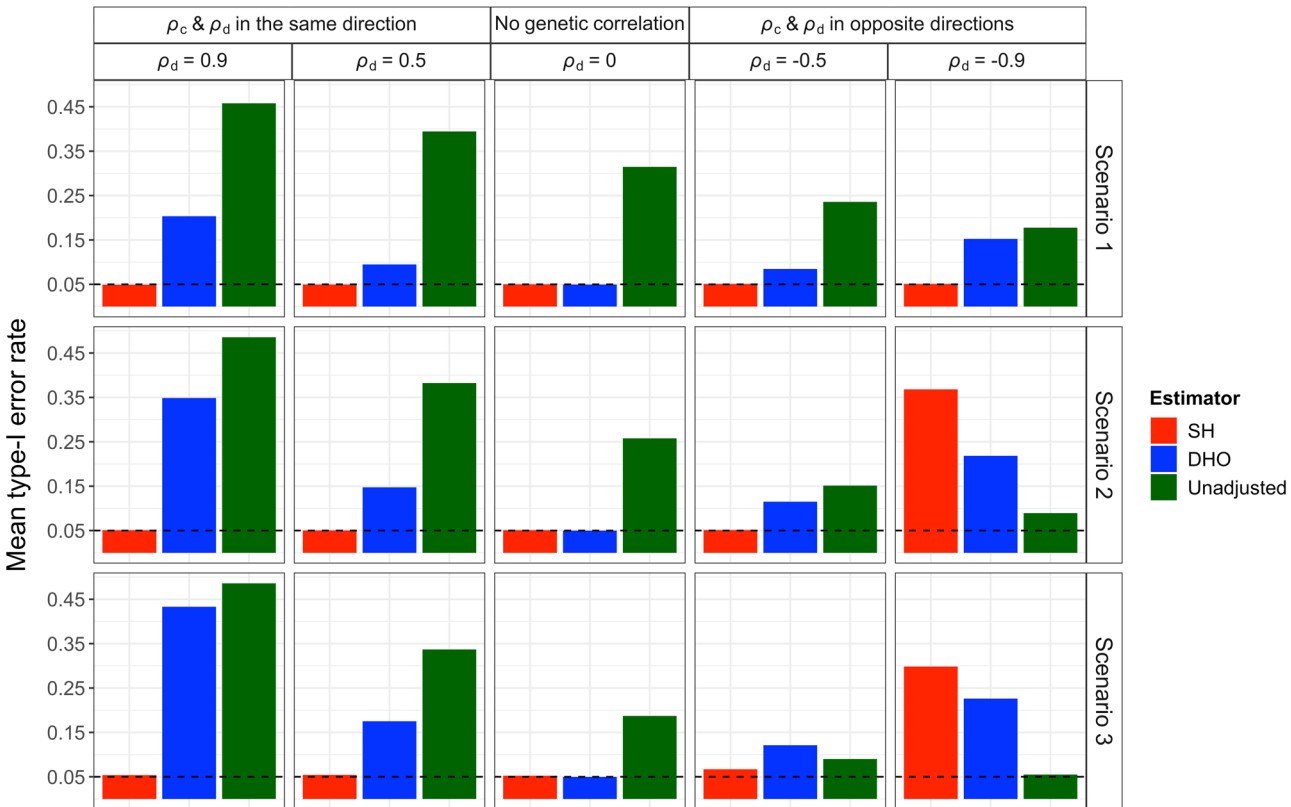

**Fig. 2 Means of type-1 error rates (averaged over SNPs affecting *I*) for the unadjusted and adjusted estimators using the Slope-Hunter (SH) method and 'Hedges-Olkin' of the Dudbridge et al. (DHO) method.** Estimates of association coefficients for 10,000 independent SNPs with a quantitative trait *I*, and a quantitative outcome (*P*) conditional on *I* are simulated assuming an underlying four-component model for effect-size distribution, where SNPs could have direct effects on *I* only ($G_I$), both traits ($G_{IP}$), *P* only or on neither traits, such that $G_I$ explains larger (Scenario 1), equal (Scenario 2), or lower (Scenario 3) proportions of the variation in *I* compared with the $G_{IP}$ variants (Methods). The direct effects on *I* and *P* could be correlated ($\rho_d \neq 0$) in the same direction as the correlation due to the confounding effect ($\rho_c$), uncorrelated ($\rho_d = 0$), or correlated in the opposite direction of the correlation due to the confounding effect ($\rho_d = 0$). The means of type-1 error rates across the $G_I$ variants over 1000 simulations are depicted for each estimator. The nominal level of the type-1 error rate (0.05) is represented by the horizontal dashed black lines. Source data are provided as a Source Data file.

variants do not suffer from the collider bias, due to the lack of effect on incidence (*I*), averaging over all variants might be misleading. Among the SNPs affecting *I*, for which there is a collider bias, the type-1 error is inflated for the unadjusted analysis, ranging from 0.06 to 0.49, and for the DHO estimator ranging from 0.09 to 0.43 in presence of genetic correlation while our procedure consistently achieves the correct rate even under genetic correlation. Under positive genetic correlation between incidence and outcome, the type-1 error rates increase for all analyses, but are consistently at the nominal level for Slope-Hunter. Similar results are obtained under moderate negative genetic correlation ($\rho_d = -0.5$). When there is strong negative genetic correlation ($\rho_d = -0.9$) and the SNPs affecting only incidence ($G_I$) do not explain larger proportion of variation in *I* than the SNPs affecting both incidence and outcome ($G_{IP}$), our approach has increased type-1 error compared with the unadjusted analysis. Figure 2 shows the mean type-1 error rates averaged only over the SNPs affecting *I* under various scenarios (Methods). The family-wise error rates follow the same pattern with more pronounced results. Some individual SNPs could have notably high type-1 error under the unadjusted analysis but is substantially reduced using our approach, and at or close to the nominal level when the genetic correlation is not strongly negative (Supplementary Tables 1–3).

The Slope-Hunter method outperforms the unadjusted and adjusted analyses using DHO method even when there are fewer SNPs with effects only on *I* ($G_I$) in relation to the total number of

SNPs affecting *I* (Scenarios S1 and S2, see "Methods"). Under scenario S1 where $G_I$ SNPs explain larger proportion of variation in *I* than $G_{IP}$ SNPs, our method consistently achieves the lowest type-1 error, the nominal level, outperforming the unadjusted and the DHO-adjusted analyses (Supplementary Table 4). Under scenario S2 where $G_I$ SNPs explain equal proportion of variation in *I* as $G_{IP}$ SNPs, our method provides the correct type-1 error rate when the genetic correlation is not negative (Supplementary Table 5). Under scenario S3 (Method), the type-1 error for the unadjusted and the adjusted analysis using the DHO method is far larger than the nominal level, but achieves the correct level under our approach (Supplementary Table 6). In scenario S4, the largest number of similar individual-SNP ratios for SNP-outcome to SNP-incidence associations comes from the $G_{IP}$ SNPs violating the ZEro Modal Residual Assumption (ZEMRA), see Methods. Under scenario S4, our approach provides high type-1 error, however still outperforms the unadjusted analysis and adjusted analysis using the DHO method, when the genetic correlation due to the direct effects and collider bias effects ($\rho_d$ and $\rho_c$, respectively) are in the same direction. When $\rho_d$ and $\rho_c$ are in opposite directions, both adjustment methods provide high type-1 error rates (Supplementary Table 7).

Figure 3 shows means of power, when averaged over the SNPs affecting *I*, for the same simulations under scenarios 1–3 (Methods). There are small to moderate drops in power for all adjusted analyses compared with the unadjusted analysis except under strong positive correlation at which our approach has a

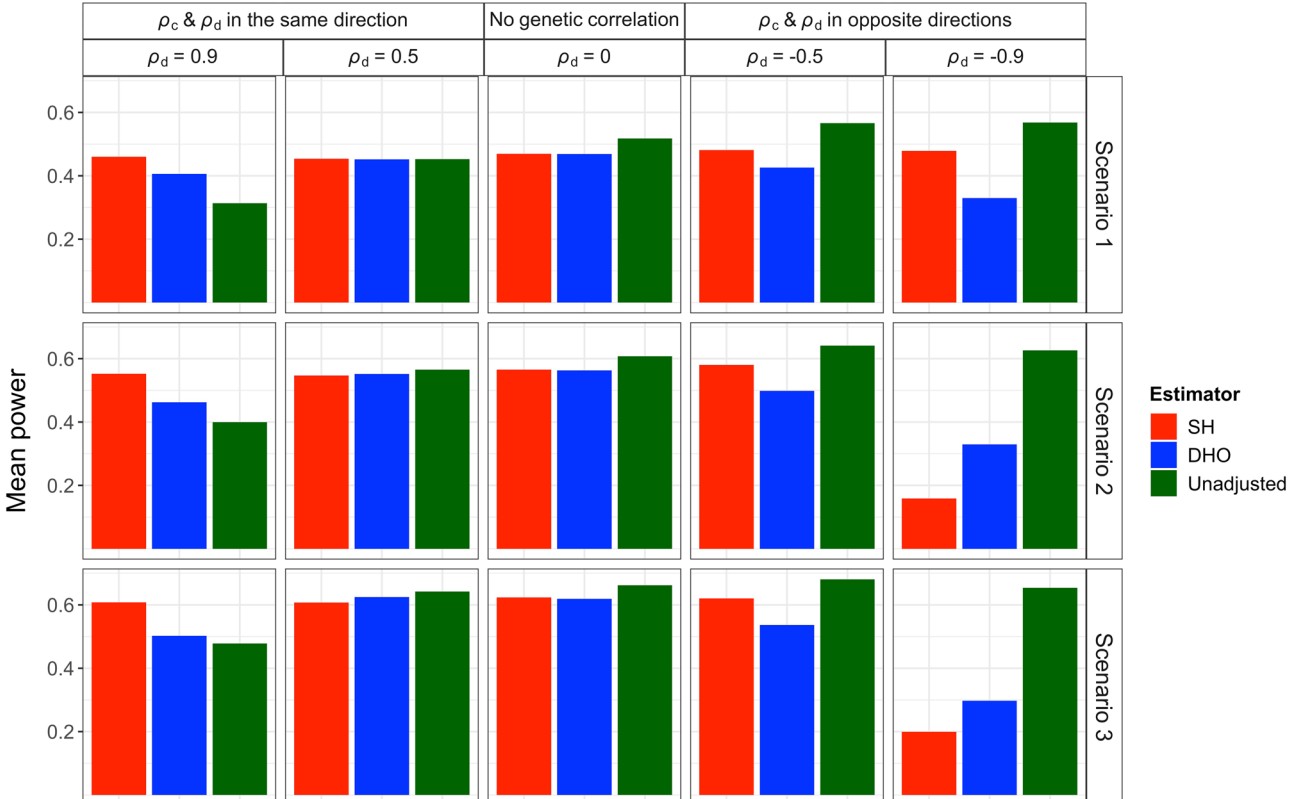

**Fig. 3 Means of power rates (averaged over SNPs affecting $I$) for the unadjusted and adjusted estimators using the Slope-Hunter (SH) method and 'Hedges-Olkin' of the Dudbridge et al. (DHO) method.** Estimates of association coefficients for 10,000 independent SNPs with a quantitative trait $I$, and a quantitative outcome ($P$) conditional on $I$ are simulated assuming an underlying four-component model for effect-size distribution, where SNPs could have direct effects on $I$ only ($G_I$), both traits ($G_{IP}$), $P$ only or on neither traits, such that $G_I$ explains larger (Scenario 1), equal (Scenario 2), or lower (Scenario 3) proportions of the variation in $I$ compared with the $G_{IP}$ variants (Methods). The direct effects on $I$ and $P$ could be correlated ($\rho_d \neq 0$) in the same direction as the correlation due to the confounding effect ($\rho_c$), uncorrelated ($\rho_d = 0$), or correlated in the opposite direction of the correlation due to the confounding effect. The means of power rates across all variants affecting $I$ and $P$ (i.e., $G_I$ and $G_{IP}$) over 1000 simulations are depicted for each estimator. Source data are provided as a Source Data file.

substantial increase. The Slope-Hunter method consistently achieves higher power than the DHO method at all levels of genetic correlation, except under strong negative correlation. For some individual SNPs, the power can be very low under the unadjusted analysis, but is substantially increased under both DHO and Slope-Hunter adjusted analyses, with the greatest increase mostly under the Slope-Hunter method (Supplementary Tables 1–3). On the other hand, some individual SNPs had a greater gain in power under the unadjusted analysis than the adjusted analyses, but this should be offset against the inflated type-1 error. Our procedure consistently yields the lowest absolute bias and lowest mean square error compared to unadjusted and adjusted analyses using alternatives at all levels of genetic correlation, except at strong negative correlation ($\rho_d = -0.9$) under scenario 2 and scenario 3. Similar results are obtained under scenarios S1-S3 (Supplementary Tables 4–6). Under the scenario S4 (where ZEMRA is violated), our approach has comparable absolute bias and mean square error rate to other analyses when $\rho_d$ and $\rho_c$ are in the same direction, but greater absolute bias and mean square when $\rho_d$ and $\rho_c$ are in opposite directions (Supplementary Table 7).

Figure 4 shows the correction factors estimated using the Slope-Hunter and DHO methods under scenarios 1–3 at different levels of genetic correlations. Under all the main scenarios (1–3) at all levels of genetic correlations, Slope-Hunter consistently provides unbiased and precise estimates of the correction factor, except under strong negative correlation ($\rho_d = -0.9$) in scenario 2

and scenario 3, whereas the DHO provides unbiased estimates only when there is no genetic correlation. Under scenario S1 and scenario S3, Slope-Hunter provides unbiased estimates for the correction factor at all genetic correlation levels, whereas the DHO method provides biased estimates in the presence of correlation. For scenario S2, Slope-Hunter provides unbiased estimates only under non-negative genetic correlations. For scenario S4, where the ZEMRA assumption is violated (Methods), both the Slope-Hunter and the DHO methods provide biased estimates (Supplementary Table 8).

Supplementary Table 9 shows type-1 error rates and estimated adjustment factors under scenario 2 (with balanced cluster sizes and balanced proportions of explained variations in $I$, see "Methods") at different $p$-value thresholds in the $z$-test for the Slope-Hunter SNP selections. Results were compared with the naive conditional analyses and with DHO adjustments. Results of the Slope-Hunter method were similar at different thresholds, producing the lowest error, mostly achieving the nominal level, providing unbiased estimates of collider bias adjustment factors. A large $p$-value threshold ($p < 0.1$) was the exception, particularly when SNP selection was performed using an independent dataset associated with $I$ from the one used for bias adjustment, where the error was increased and the estimate of adjustment factor was both biased and imprecise.

**BMI-adjusted fasting insulin.** A GWAS meta-analysis for up to 30,825 non-diabetic individuals with European ancestry identified

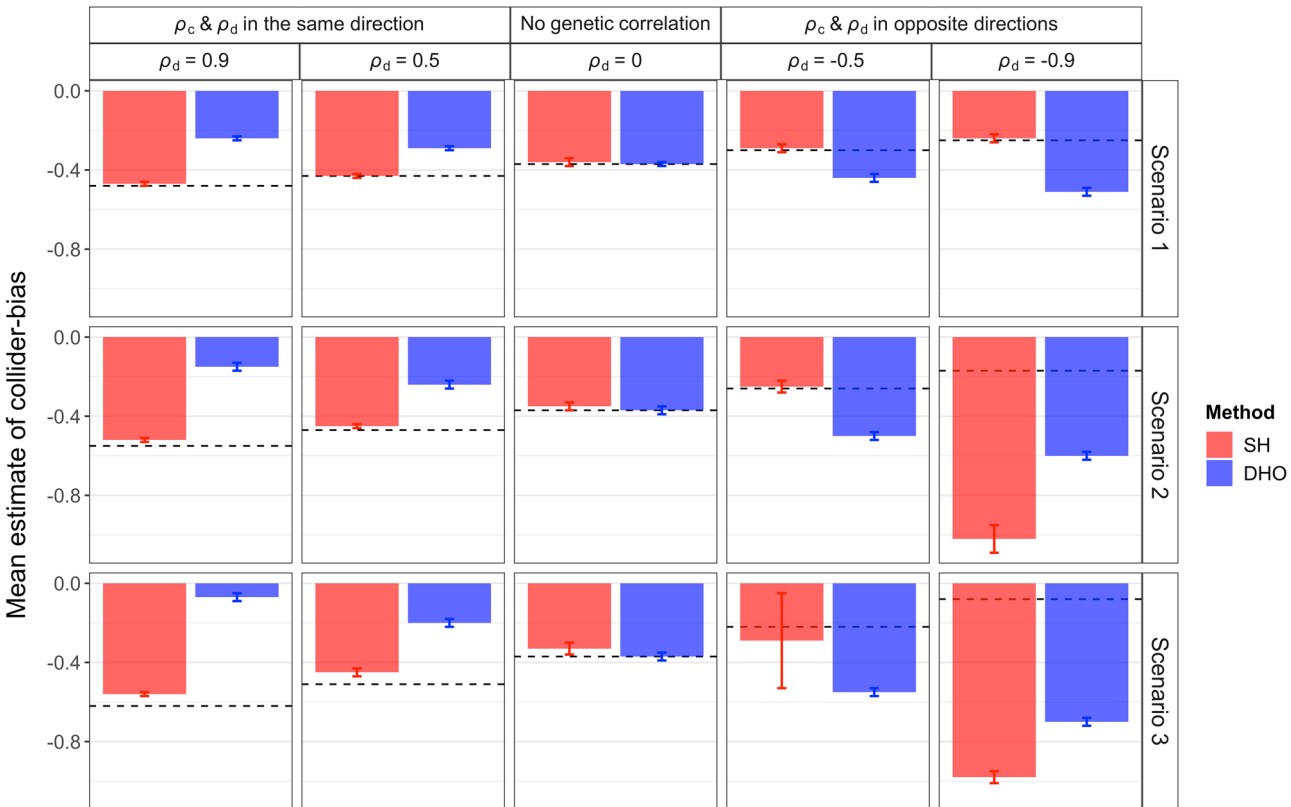

**Fig. 4 Adjustment factors (means ± SD) for the collider bias that are estimated using the Slope-Hunter (SH) and the 'Hedges-Olkin' estimator of the Dudbridge et al.** (DHO)[7] methods under different simulated scenarios. The means and the error bars are estimated from $n = 1000$ independent simulations. The true collider bias induced under different scenarios are depicted by the horizontal dashed black lines. In these simulation studies, estimates of association coefficients for 10,000 independent SNPs with a quantitative trait $I$, and a quantitative outcome ($P$) conditional on $I$ are simulated assuming an underlying four-component model for effect-size distribution, where SNPs could have direct effects on $I$ only ($G_I$), both traits ($G_{IP}$), $P$ only or on neither traits, such that $G_I$ explains larger (Scenario 1), equal (Scenario 2), or lower (Scenario 3) proportions of the variation in $I$ compared with the $G_{IP}$ variants (Methods). The direct effects on $I$ and $P$ could be correlated ($\rho_d \neq 0$) in the same direction as the correlation due to the collider-bias effect ($\rho_c$), uncorrelated ($\rho_d = 0$), or correlated in the opposite direction of the correlation due to the collider-bias effect. Source data are provided as a Source Data file.

associations of five variants at known GWAS regions with BMI-adjusted fasting blood insulin (FI) at exome-wide significance ($p < 5e^{-7}$)[1]. All of these regions, whose mapped genes are *GCKR*, *C2orf16*, *GPNI*, *COBLLI* and *PPARG*, were associated with BMI, except for the SNP rs3749147 in *GPNI* (Wald test $P = 1.3 \times 10^{-6}$)[16]. The risk alleles of the SNPs rs7607980 (the *COBLLI* gene) and rs1801282 (the *PPARG* gene) were associated with decreased BMI but increased FI. These apparently paradoxical associations could arise from collider bias, given their associations with BMI (Table 1). The variant rs1421085 (gene *FTO*) had the strongest association with BMI, but appeared to be not associated with FI. The lack of association with FI could also arise from collider bias, given the strong association with BMI.

We analysed the GWAS summary statistics for BMI and fasting blood insulin conditional on BMI (FI) using the Slope-Hunter method and existing alternative methods. The adjustment factor estimated using the Slope-Hunter method was $-0.317$ (95% CI: $-0.417$ to $-0.218$ based on a standard error of 0.051 estimated using 10,000 bootstrap samples), whereas the DHO method gave an estimate of $-0.118$, which slightly changed to $-0.113$ (95% CI: $-0.151$ to $-0.084$) when corrected for regression dilution using DSIMEX[7]. The adjustment factors obtained by all adjustment methods are negative implying that there are common causes of BMI and fasting blood insulin of concordant net directions of effect. Under the adjusted analyses, the two apparently paradoxical associations of *COBLL1* and

*PPARG* were attenuated towards the null, with greater attenuations under the Slope-Hunter method (Table 1). Our adjustment revealed an association for the *FTO* gene with fasting blood insulin in a direction that is concordant with the direction of its association with BMI. The same direction of association was obtained using the DHO method, but the estimated coefficient was closer to the null. Applying the mtCOJO method[15] to adjust for the collider bias in this conditional analysis requires the marginal summary-level GWAS data for FI, i.e. FI GWAS that is not adjusted for BMI, which is not available from the considered studies. We have compared the adjustments obtained by the Slope-Hunter and the DHO methods with the adjusted estimates obtained using the Generalised Summary-data-based Mendelian Randomisation (GSMR) method, the core procedure of the mtCOJO method, which could be used to estimate the collider bias and, through that, to derive an adjustment factor for the collider bias correction using the considered summary-level data. This enabled comparison of the adjustments of Slope-Hunter with the alternatives for exactly the same variants considered in our analyses (Supplementary Table 10).

**Breast cancer mortality.** We analysed summary statistics of a recent GWAS study for breast cancer[17] and of a case-only study for breast cancer mortality[5]. Table 2 shows the associations of three variants with disease susceptibility, their unadjusted associations with the prognosis, and their adjusted results using

**Table 1 Associations of six genes with BMI-conditioned fasting blood insulin levels (FI): five genes are exome-wide significant ($p^b < 5e^{-7}$); one gene (*FTO*) is strongly associated with BMI, but not associated with FI.**

| Variant | Gene | Alleles[a] | Association with BMI $\hat{\beta}$(SE) | Association with fasting blood insulin | | | Adjusted using SH | | Adjusted using DHO | |
| --- | --- | --- | --- | --- | --- | --- | --- | --- | --- | --- |
| | | | | Unadjusted $\hat{\beta}$(SE) | $p^b$ | $p^c$ | $\hat{\beta}$(SE) | $p^d$ | $\hat{\beta}$(SE) | $p^d$ |
| rs1260326 | GCKR | T/C | −0.011 (0.001) | −0.023 (0.004) | $8.08e^{-11}$ | $2.72e^{-9}$ | −0.026 (0.004) | $1.4e^{-11}$ | −0.023 (0.004) | $3.73e^{-10}$ |
| rs1919128 | C2orf16 | A/G | 0.009 (0.001) | 0.022 (0.004) | $1.46e^{-8}$ | $1.85e^{-7}$ | 0.025 (0.004) | $4.0e^{-9}$ | 0.023 (0.004) | $4.32e^{-8}$ |
| rs3749147 | GPN1 | A/G | −0.009 (0.002) | −0.022 (0.005) | $1.22e^{-7}$ | $1.39e^{-6}$ | −0.025 (0.004) | $5.4e^{-8}$ | −0.023 (0.005) | $4.07e^{-7}$ |
| rs7607980 | COBLL1 | T/C | −0.015 (0.003) | 0.030 (0.006) | $6.71e^{-8}$ | $2.78e^{-7}$ | 0.025 (0.006) | $1.8e^{-5}$ | 0.028 (0.006) | $1.38e^{-6}$ |
| rs1801282 | PPARG | C/G | −0.017 (0.003) | 0.025 (0.004) | $1.25e^{-7}$ | $1.14e^{-6}$ | 0.019 (0.005) | $1.6e^{-4}$ | 0.023 (0.005) | $7.99e^{-6}$ |
| rs1421085 | FTO | T/C | −0.078 (0.002) | 0.001 (0.004) | 0.555 | 0.871 | −0.024 (0.004) | $1.2e^{-10}$ | −0.006 (0.004) | 0.091 |

BMI body mass index, $\hat{\beta}$ regression coefficient estimate, SE standard error, SH Slope-Hunter method, DHO Dudbridge's method[7].
[a] Alleles are reported as the 'effect'/'other' allele.
[b] Unadjusted sample-size Z-score weighted p-value, reported by Mahajan et al.[1].
[c] Two-sided Wald test p-value, unadjusted for collider bias, based on the estimates reported by Mahajan et al.[1].
[d] Two-sided Wald test p-value from adjusted analyses.
The lead variants in COBLL1 and PPARG, rs7607980 and rs1801282, respectively, associate with decreased BMI but with increased BMI-conditioned fasting insulin levels achieving exome-wide significance in the unadjusted analysis. Our adjusted analysis suggests that these apparently paradoxical associations could be due to the collider bias. Therefore, their associations were attenuated towards the null and became not significant under our adjusted analysis. The lead variant in FTO, rs1421085, is not associated with fasting insulin levels under the unadjusted analysis. Our adjusted analysis suggests that the lack of association could be due to collider bias that could have dramatic effect given the strong association with BMI. Our corrected estimate for this SNP suggested a genome-wide significant association with fasting blood insulin levels in the same direction as its association with BMI.

Slope-Hunter and DHO methods. One variant (rs370332736) has the lowest *p*-value under the naive conditional (unadjusted) analysis, and the other two have the strongest associations with the disease susceptibility. The estimated adjustment factor obtained by the Slope-Hunter was −0.242 (95% CI: −0.363 to −0.121 based on a standard error of 0.062 estimated using 10, 000 bootstrap samples), whereas it was −0.053 and −0.013 (95% CI: −0.014 to −0.012) using the DHO and DSIMEX methods, respectively.

The risk allele of rs35054928 (gene *FGFR2*) associated with reduced risk of the incidence (odds ratio 0.76; 95% CI: 0.73 to 0.79; Wald test $p = 1.6e^{-43}$) suggesting a strong protective role against the risk of breast cancer, but was associated, although without statistical significance, with increased breast cancer mortality. This result could arise from collider bias, given the strong association with incidence. Our adjustment approach changed the direction of association for this variant with prognosis (Table 2). Results of our adjusted analysis for the SNP rs35850695 (gene *Tox3*) showed more pronounced association with prognosis ($\hat{\beta} = 0.07$; 95% CI: 0.035 to 0.105; Wald test $p = 3e^{-4}$) compared with the unadjusted association ($\hat{\beta} = 0.01$; 95% CI: −0.029 to 0.049; $p = 0.614$). The adjusted associations using the DHO were not substantially different from the results obtained from the unadjusted analysis, given the very small magnitude of the adjustment factor. Differences between DHO and slope-hunter could be explained by potential violations to the InCLUDE assumption, a key assumption for the DHO method, as breast cancer incidence and mortality are likely to share genetic pathways that may result in correlated effects on both incidence and prognosis.

## Discussion

Conditional analyses of genetic associations with an outcome, such as prognosis, subsequent disease events, severity and survival time, are increasingly motivated by many large collections of GWAS for disease cases. Such case-only studies are liable to collider bias, whereby independent causes of the incidence become correlated when selecting only on cases, inducing bias in the analysis of outcome. We have proposed an approach that overcomes a major disadvantage of previous methods, and showed that it provides unbiased estimates of SNP-outcome associations in a variety of situations, including in the presence of genetic correlations between I (e.g. incidence) and outcome (e.g. prognosis). Our approach aims to identify the set of SNPs with effects only on I and uses it to estimate and adjust for the collider bias induced by the confounder effects. Our approach is robust against the violation of the InCLUDE assumption that is required by other methods[7]. Our analytic approach assumes the analysed SNPs are independent, do not interact with the confounders, have linear effects on I and outcome, and have no interaction with I. Moreover, it requires the ZEro Modal Residual Assumption (ZEMRA) that resembles the ZEMPA assumption for the MR analysis, but with respect to the residuals ($e_G = \beta'_{GP} - b_1 \beta_{GI}$) rather than pleiotropy. The ZEMRA, like ZEMPA, is generally not a testable assumption since the true clusters of all SNPs are usually unknown. When its assumptions are satisfied, the Slope-Hunter method can maintain excellent trade-off between type-1 error rates and power, and produce lower mean square error compared to the other methods, even in the presence of genetic correlations.

We ran extensive simulations with various levels of correlations between genetic effects on I and outcome under different scenarios. The simulation studies showed that the Slope-Hunter method provided unbiased estimates of the true collider bias, achieved the minimum type-1 error rates, minimum mean square error, with comparable power on average and considerably higher

**Table 2 Associations of three SNPs with breast cancer mortality: one SNP (rs370332736) is exome-wide significant ($p < 5e^{-7}$); the other two are strongly associated with breast cancer incidence, but not appeared to be associated with breast cancer mortality under the unadjusted analyses.**

| | | | | Association with breast cancer risk | Association with breast cancer mortality | | | | | |
| | | | | | Unadjusted | | Adjusted using SH | | Adjusted using DHO | |
| Variant | Chromosome | Gene | Alleles[a] | $\hat{\beta}$(SE) | $\hat{\beta}$(SE) | p | $\hat{\beta}$(SE) | p | $\hat{\beta}$(SE) | p |
|---|---|---|---|---|---|---|---|---|---|---|
| rs370332736 | 6 | – | A/AACTT | 0.05 (0.03) | 0.15 (0.03) | $2.5e^{-7}$ | 0.17 (0.03) | $6.7e^{-8}$ | 0.16 (0.03) | $1.6e^{-7}$ |
| rs35054928 | 10 | FGFR2 | G/GC | −0.27 (0.02) | 0.03 (0.02) | 0.085 | −0.04 (0.017) | 0.034 | 0.01 (0.02) | 0.394 |
| rs35850695 | 16 | Tox3 | A/G | 0.23 (0.02) | 0.01 (0.02) | 0.614 | 0.07 (0.018) | $3.0e^{-4}$ | 0.02 (0.02) | 0.226 |

$\hat{\beta}$ regression coefficient estimate, *SE* standard error, *SH* Slope-Hunter method, *DHO* Dudbridge's method[7].
[a]Alleles are reported as the 'effect/other' allele.
*p*-values are reported from two-sided Wald tests.

power for some individual SNPs, compared with the unadjusted analysis and alternative adjustment methods even under the presence of genetic correlations. All methods had worse type-1 error rates as the proportion of variation in $I$ explained by SNPs affecting $I$ only reduced. However, the Slope-Hunter method had better type-1 error than the alternatives when the ZEMRA holds. When genetic direct effects on $I$ and $P$ were strongly correlated in the opposite direction to the correlation due to collider bias, the Slope-Hunter method gave higher type-1 error rates under scenarios with lower effect-sizes of the SNPs affecting $I$ only ($G_I$). However this situation, if not implausible, is arguably less likely to occur, particularly with strong negative genetic correlation[2].

Our analysis of BMI-adjusted fasting blood insulin (FI) suggests that apparently paradoxical associations of the strong risk loci *COBLL1* and *PPARG* with increased insulin levels may be partly due to collider bias, and that these associations have been attenuated towards the null after adjustment. It has been suggested that risk alleles of the *COBLL1* and *PPARG* genes have considerable associations with BMI[16], and this could lead to biased association when conditioning on BMI, due to collider bias[1]. The association of another strong BMI risk loci (*FTO* gene) with the outcome (FI) after correction showed a strong association in the same direction as its association with BMI. Our findings suggest that the common causes of BMI and insulin levels, the source of the collider bias, have effects on both traits with concordant directions. We have presented the results of six variants that were either strongly associated with BMI or associated with the outcome before correction. The concordant directions, identified by our analysis, are in line with the observed association between insulin resistance and obesity[18,19] and agrees with the adjustment factor estimated using alternative methods[7]. In another study, we analysed the breast cancer mortality in a large case-only study. In this study, the slope-Hunter method estimated that common causes of breast cancer and all-causes mortality of breast cancer act in concordant directions. When correcting for the collider bias induced from these common causes, the associations of two risk loci (*FGFR2* and *Tox3* genes) were either flipped to the intuitive direction or became more pronounced in the intuitive direction, compared with their unadjusted associations.

The Slope-Hunter method requires user choice for the input parameter ($\lambda$) that controls exclusion of SNPs with no effect on $I$. Large values of $\lambda$ (closer to 1) can lead to inclusion of more SNPs in the analysis which may improve clustering due to the potential increase in number of SNPs affecting $I$, i.e. results in larger size of the identified cluster ($G_I$). However, including too many SNPs may also result in including a fraction of SNPs with no effects on $I$ that may obscure the pattern of the $G_I$ cluster. Although our

model could be incorrect, approximating the underlying model-based cluster, under excessive inclusion of null SNPs, our simulation studies suggest that the estimate of correction factor ($\hat{b}_1$) has no bias, even under a relatively large threshold ($\lambda = 0.001$). However, a user should perform the Slope-Hunter analysis at different values of $\lambda$ to examine the sensitivity of their data to the change in the $\lambda$ parameter.

The main idea of our procedure can be adopted in future in the context of the MR analysis using a large number of genetic variants including invalid instruments, particularly for experiments in which effects of instruments on exposure and outcome are correlated[20]. This potential direction may be beneficial in robustly estimating causal effects, checking violation of MR assumptions, providing probabilistic identification of the valid instruments, and detecting pleiotropy in a given problem. A few methods, e.g. the MR-mix[21] and CAUSE[22], have been recently developed with a conceptual similarity to the Slope-Hunter method in the context of MR analysis. The aim of these methods is to use mixture models with valid and potentially correlated invalid instruments to estimate causal effect of an exposure on outcome. The Slope-Hunter approach can be adapted to identify the class of SNPs that show no pleiotropy (equivalent to class $G_I$ in this context), and the class that demonstrates pleiotropy (class $G_{IP}$ in this context). This approach would likely be robust to the 'Instrument Strength Independent of Direct Effect' (InSIDE) assumption[20] but may require the ZEMPA assumption.

Our study has several strengths. It provides a framework to correct for collider bias even in the presence of genetic correlations between $I$ and $P$, i.e. it is robust against violation of the 'InCLUDE' assumption, that is required by other methods[7]. We validated our developed approach in a wide range of simulations under various scenarios with different combinations of: genetic correlations; magnitudes of genetic confounders; number of SNPs with effects only on $I$; proportions of explained variation in $I$. Our study compared the performance of our method with the unadjusted analysis and other alternative methods in terms of many statistical criteria including type-1 error, power, bias and mean squared error. Nevertheless, our study has a number of limitations. Although we have examined performance of the Slope-Hunter method in different situations, we have not examined the sensitivity to non-linearity or to interaction between confounder and variant's effects. There is not a single criterion for validity of the Slope-Hunter approach, as it will depend on how separated the classes $G_I$ and $G_{IP}$ are and whether the class $G_I$ is correctly identified. Assumptions of the Slope-Hunter method are not testable, but comparative studies with alternative methods, that have different assumptions, can give insights into performance of these methods under different situations.

We have proposed an approach for adjusting conditional genetic associations studies for collider bias even in the presence of genetic correlation between $I$ and $P$. We recommend that this approach is used in GWAS of subsequent events, e.g. for case-only studies, to minimise the bias due to conditioning on $I$. This approach is also recommended for subsequent use of GWAS results, such as in MR analyses of the effect of exposure on prognosis. All procedures described in this manuscript have been implemented into an open source R package named 'SlopeHunter'[23].

## Methods

**Model setup**. We propose an approach based on model-based clustering, as with similar approaches in MR[21,22]. We follow the structure of Qi et al.[21] but propose a different clustering method for adjustment of collider bias in GWAS analyses that requires only summary-level association statistics for a putative trait ($I$) and the outcome ($P$) conditional on $I$. Our context implies that $P$ can be a subsequent trait analysed in a case-only study, in which the incidence trait ($I$) is binary. We describe our proposed method in the context of independent SNPs. Let $(\beta_{GI}, \beta_{GP})$, $G = 1, …, M$, denote the underlying true association coefficients of the $M$ independent SNPs for the trait ($I$) and the outcome ($P$) conditional on $I$, respectively. For an individual SNP ($G$), it is assumed that $I$ and $P$ follow linear models of the forms:

$$I = \beta_{GI} G + \beta_{UI} U + \varepsilon_I,$$

$$P = \beta_{GP} G + \beta_{UP} U + \beta_{IP} I + \varepsilon_P,$$

where $U$ denotes the common causes of $I$ and $P$ (including polygenic common effects and non-genetic common factors), while $\varepsilon_I$ and $\varepsilon_P$ refer to unique causes of $I$ and $P$, respectively. We assume, without loss of generality, that $G$, $U$, $\varepsilon_I$ and $\varepsilon_P$ each have mean zero, have no interactions and are pairwise uncorrelated, and hence also $E(I) = E(P) = 0$.

These models can then be expressed with respect to $G$, $U$ and $I$ as follows:

$$E(I|G, U) = \beta_{GI} G + \beta_{UI} U, \quad (1)$$

$$E(P|G, U, I) = \beta_{GP} G + \beta_{UP} U + \beta_{IP} I. \quad (2)$$

The effect of interest is the direct SNP effect on outcome ($\beta_{GP}$), that is conditional on $I$ and confounders $U$. However in practice, we can only estimate the SNP-outcome association conditional on $I$, as all relevant confounders may not be observed:

$$E(P|G, I) = \beta'_{GP} G + \beta'_{IP} I, \quad (3)$$

where $\beta'_{GP}$ is a biased estimate of SNP effect on outcome (which is biased because conditioning on $I$ induces collider bias via $U$), whereas $\beta'_{IP}$ is a biased estimate of the causal effect of $I$ on $P$ (which is biased because of the confounding effect of $U$ rather than because of the collider bias).

Dudbridge et al.[7] showed that the biased effect ($\beta'_{GP}$) can be formulated as the true effect ($\beta_{GP}$) plus a bias that is linear in the SNP effect on $I$ ($\beta_{GI}$)[7]:

$$\beta'_{GP} = \beta_{GP} + b \beta_{GI}, \quad (4)$$

$$b = \frac{-\sigma_U^2 \beta_{UI} \beta_{UP}}{\sigma_U^2 \beta_{UI}^2 + \sigma_{\varepsilon_I}^2}, \quad (5)$$

where $\sigma_U^2$ and $\sigma_{\varepsilon_I}^2$ are variances of confounders and residual unique causes of $I$, respectively. The linear relationship between the biased effect ($\beta'_{GP}$) and the bias approximately holds for binary $I$, e.g. if $I$ represents a disease status as in case-only studies, or binary $P$ since the logistic and probit link functions are approximately linear for small effects, as typically is the case for polygenic traits[7]. The model of binary trait $I$ can be expressed as

$$logit[\Pr(I)|G] = \beta_{GI} G,$$

where $\Pr(I)$ is the probability of $I = 1$, whereas $\beta_{GI}$ represents the logarithm of odds ratio. The model of outcome ($P$) in cases only can then be expressed as:

$$E(P|G, I=1) = \beta''_{GP} G + \beta'', \quad (6)$$

where $\beta''_{GP}$ is the biased estimate of SNP effect on outcome, and $\beta''$ is the intercept.

It has been shown that the slope, $b$ in Eq. (4), could be estimated using ordinary least squares (OLS), by regressing $\beta'_{GP}$ on $\beta_{GI}$ for all SNPs assuming that[7]:

- $A_1$: The effects of SNPs on incidence ($I$) are linearly uncorrelated with their direct effects on outcome ($P$), i.e. Incidence Coefficient Linearly Uncorrelated with Direct Effect ('InCLUDE' assumption).
- $A_2$: The confounder effects—and hence $b$—are constant across all SNPs.

The estimated slope, $\hat{b}$, can then be used to obtain bias-adjusted association with the outcome for each SNP by calculating the residuals from the model in Eq.

(4) as follows:

$$\hat{\beta}_{GP} = \hat{\beta}'_{GP} - \hat{b} \hat{\beta}_{GI}. \quad (7)$$

If there are shared pathways for both $I$ and $P$ whereby the direct effects on the outcome ($P$) are correlated with effects on $I$, e.g. as shown in Fig. 1c, then the InCLUDE assumption ($A_1$) can be violated producing bias in $\hat{b}$, and hence not correcting adequately for the collider bias, see Fig. 5.

**Motivating idea**. We assume a SNP ($G$) can belong to one of four mutually exclusive clusters according to its effects on the traits $I$ and $P$:

1. $G_I$: denotes a SNP from the cluster that causes $I$ but has no direct effect on $P$ (Fig. 1a), with the following distributional assumption:

$$\beta_{G_I I} \sim N(0, \sigma_I^2), \qquad \beta_{G_I P} = 0. \quad (8a)$$

2. $G_{IP}$: denotes a SNP from the cluster that has direct effects on both $I$ and $P$ (Fig. 1b, c), with the following distributional assumption:

$$\begin{pmatrix} \beta_{G_{IP} I} \\ \beta_{G_{IP} P} \end{pmatrix} \sim N\left( \begin{pmatrix} 0 \\ 0 \end{pmatrix}, \begin{pmatrix} \sigma_I^2 & \sigma_{IP} \\ \sigma_{IP} & \sigma_P^2 \end{pmatrix} \right) \quad (8b)$$

3. $G_P$: denotes a SNP from the cluster that has direct effect on $P$, but has no relationship with $I$ (Fig. 1d), with the following distributional assumption:

$$\beta_{G_P I} = 0, \qquad \beta_{G_P P} \sim N(0, \sigma_P^2). \quad (8c)$$

4. $G_{..}$: denotes a SNP from the cluster that relates to neither $I$ nor $P$ (Fig. 1e), with the following distributional assumption:

$$\beta_{G_. I} = 0, \qquad \beta_{G_. P} = 0. \quad (8d)$$

The SNPs in the first two clusters ($G_I$ and $G_{IP}$) have non-zero bias terms, $\beta'_{GP} - \beta_{GP} \neq 0$, whose magnitude is proportional to their effects on $I$, see Eq. (4). SNPs of the second cluster ($G_{IP}$) have potential correlated effects on $I$ and $P$. This allows violation of the InCLUDE assumption ($A_1$) formulated by Dudbridge et al.[7], as we allow $\sigma_{IP} \neq 0$ (Eq. (8b)). The SNPs in the third ($G_P$) and fourth ($G_{..}$) clusters are not associated with $I$, hence they do not suffer bias, i.e. $\beta'_{GP} = \beta_{GP}$. Consequently, we reformulate Eq. (4) as follows:

$$\beta'_{GP} = \begin{cases} b_1 \beta_{GI}, & \text{for variants causing } I \text{ only } (G_I) \\ \beta_{GP} + b_{2G} \beta_{GI}, & \text{for variants causing } I \text{ \& } P \text{ } (G_{IP}) \\ \beta_{GP}, & \text{for variants causing } P \text{ only } (G_P) \\ 0, & \text{for variants causing neither } I \text{ nor } P \text{ } (G_{..}) \end{cases} \quad (9)$$

Instead of regressing $\hat{\beta}'_{GP}$ on $\hat{\beta}_{GI}$ for all SNPs, as implemented in alternative methods[7], we propose modelling the bivariate distribution of the effect-sizes ($\beta'_{GP}$ and $\beta_{GI}$) using a Gaussian model-based clustering technique from which the cluster of $G_I$ SNPs can be identified, and then used for estimating the the correction factor ($b_1$). This requires the proportional relationship between $\beta'_{GP}$ and $\beta_{GI}$ to hold only for a fraction of the genetic variants, that is $G_I$, rather than across all genetic variants being analysed. The estimated correction factor ($\hat{b}_1$) can then be used to correct bias for all SNPs, by substituting $\hat{b}$ by $b_1$ in Eq. (7) assuming the confounder effects are constant across all SNPs (assumption $A_2$) under which $b_1 = b_{2G}$ for all $G = G_{IP}$ in Eq. (9).

**Collider-bias correction using model-based clustering**. Assuming the confounder effects are constant across all SNPs, the distributions of the SNP-$I$ and SNP-$P$ associations can be written under the proposed model in the form of

$$\begin{pmatrix} \beta_{GI} \\ \beta'_{GP} \end{pmatrix} \sim \pi_1 N\left( \underline{0}, \begin{bmatrix} \sigma_I^2 & b_1 \sigma_I^2 \\ b_1 \sigma_I^2 & b_1^2 \sigma_I^2 \end{bmatrix} \right) + \pi_2 N\left( \underline{0}, \begin{bmatrix} \sigma_I^2 & b_1 \sigma_I^2 + \sigma_{IP} \\ b_1 \sigma_I^2 + \sigma_{IP} & b_1^2 \sigma_I^2 + \sigma_P^2 + 2b_1 \sigma_{IP} \end{bmatrix} \right)$$
$$+ \pi_3 \begin{pmatrix} \eta_0 \\ N(0, \sigma_P^2) \end{pmatrix} + \pi_4 \begin{pmatrix} \eta_0 \\ \eta_0 \end{pmatrix}, \quad (10)$$

where $\pi_1$, $\pi_2$, $\pi_3$ and $\pi_4$ denote the probabilities that a SNP belongs to the clusters $G_I$, $G_{IP}$, $G_P$ and $G_{..}$, defined in Eqs. (8a)–(8d), respectively, with $\sum_{k=1}^4 \pi_k = 1$, whereas $\underline{0}$ is a $2 \times 1$ zero-vector and $\eta_0$ is the probability point mass at 0. The latter two components in the model, shown in Eq. (10), represent clusters ($G_P$ and $G_{..}$) that do not affect $I$, hence do not suffer from the collider bias, and are then uninformative for our analysis. Since the SNP-I associations are observed with no collider bias, in our context, then the SNPs $G_P$ and $G_{..}$ could be effectively identified by employing a $p$-value threshold in the study associated with $I$ to exclude SNPs that are not associated with the trait ($I$).

From GWAS of $I$ and the conditional analysis of $P$ on $I$, we obtain estimates ($\hat{\beta}_{GI}$) and biased estimates ($\hat{\beta}'_{GP}$), respectively, where one can assume $\hat{\beta}_{GI} \sim N(\beta_{GI}, s_I^2)$, $\hat{\beta}'_{GP} \sim N(\beta'_{GP}, s_P'^2)$, with estimated standard errors $s_I$ and $s_P'$. The biased standard error ($s_P'$) can be expressed as a function of the collider bias correction factor ($b_1$) as $s_P' = \sqrt{s_P^2 + b_1^2 s_I^2 + 2b_1 \sigma_{IP}}$ (see Eq. (9)), where $s_I$ and $s_P$ are the standard errors of

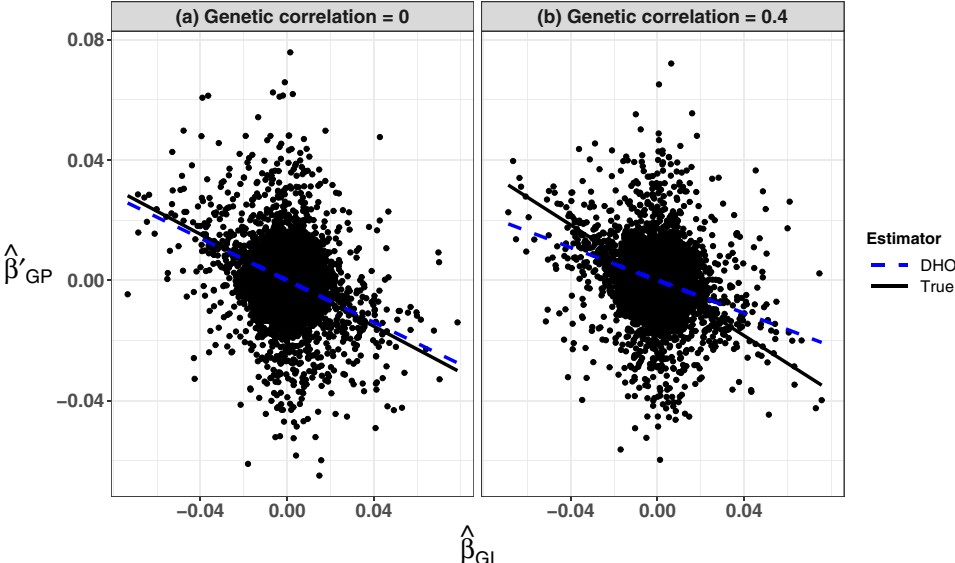

**Fig. 5 Scatter plots for simulated estimates of SNP association with a quantitative trait (I), $\hat{\beta}_{GI}$, and a quantitative outcome (P) conditional on I, $\hat{\beta}'_{GP}$.**
The estimates are simulated for 10,000 independent SNPs from a dataset of 20,000 individuals, with: (**a**) no genetic correlation between SNP effects on I and P; (**b**) correlated genetic effects on I and P (correlation coefficient = 0.4). In both analyses, (**a**) and (**b**), the SNP associations are simulated under a hypothesised four-component model for effect-size distribution in which 5% of SNPs have effects on I only, 5% on P only, 5% on both and 85% on neither. The heritability of I and P is 50% and the non-genetic common factors explain 40% of variation in both I and P. The analyses in both (**a**) and (**b**) induced collider bias due to the common causes of I and P, including common polygenic effect as well as non-genetic common factors. The true collider biases are represented by slopes of the black solid lines, which are −0.383 and −0.460 in (**a**) and (**b**), respectively, while the estimated correction factors using the `Hedges-Olkin' estimator of the Dudbridge et al. (DHO) method[7] are represented by slopes of the blue dashed lines, which are −0.349 and −0.273 in (**a**) and (**b**), respectively. The analysis depicted in (**b**) illustrates potential inadequate correction using the DHO method when the `InCLUDE' assumption (Index Coefficient Linearly Uncorrelated with Direct Effect) is violated. Source data are provided as a Source Data file.

the direct effects on I and P, respectively, and $\sigma_{IP}$ denotes the covariance between effects on both traits.

In the following, we propose a correction factor estimation procedure that is computationally simple and relies on model-based clustering to identify the cluster ($G_I$) with a proportional relationship between the SNP-I and SNP-P associations. Our procedure solves a clustering problem in 2-dimensional space using a bivariate Gaussian mixture model of the effect-size distributions. The true correction factor ($b_1$) maximises the density of points lying on, or scattered 'closely' around, the line $\hat{\beta}'_{GP} = b_1 \hat{\beta}_{GI}$, i.e. points for which there is a proportional relationship between SNP-I and SNP-P associations (the key characteristic underlying the true cluster of SNPs, $G_I$, affecting only I), that are covered by the distribution

$$N\left(\underline{0}, \left[ s_I^2 b_1 \ \ s_I^2 b_1 \ \ s_I^2 b_1^2 \ \ s_I^2 \right]\right). \tag{11}$$

Our approach uses this property to estimate the correction factor ($b_1$) and then use this to derive the bias-adjusted associations.

The Slope-Hunter estimation procedure is presented in Box 1. Since the correction factor, $b_1$, should be estimated using a set of independent SNPs ($G$), GWAS are first pruned by linkage disequilibrium (LD). The procedure starts by using the pruned GWAS statistics for I to calculate $p$-values of the SNP-I associations (line 2), and retain only the SNPs associated with I whose $p$-values are less than a threshold $\lambda$ (line 4). The distributions of the observed associations for the variants affecting I, for which there is a collider bias are addressed as follows:

$$\begin{pmatrix} \hat{\beta}_{GI} \\ \hat{\beta}'_{GP} \end{pmatrix} \sim \pi_1^* N\left( \underline{0}, \begin{bmatrix} s_I^2 & b_1 s_I^2 \\ b_1 s_I^2 & b_1^2 s_I^2 \end{bmatrix} \right)$$
$$+ (1 - \pi_1^*) N\left( \underline{0}, \begin{bmatrix} s_I^2 & b_1 s_I^2 + \sigma_{IP} \\ b_1 s_I^2 + \sigma_{IP} & b_1^2 s_I^2 + \sigma_P^2 + 2 b_1 \sigma_{IP} \end{bmatrix} \right), \tag{12}$$

where $\pi_1^*$ represents the probability that a SNP $G$ belongs to the cluster ($G_I$) affecting only I. We use the EM algorithm[24] to estimate the unknown parameters, $b_1$, $\sigma_{IP}$ and $\pi_1^*$ (line 5). We use the Bootstrap estimation technique[25] to estimate standard error of the correction factor, $s(\hat{b}_1)$ (line 6).

The estimated correction factor ($\hat{b}_1$) is then used to derive the bias-adjusted associations for all SNPs as the residual of their biased association from the line $\hat{\beta}'_{GP} = \hat{b}_1 \hat{\beta}_{GI}$ (line 8):

$$\hat{\beta}_{GP} = \hat{\beta}'_{GP} - \hat{b}_1 \hat{\beta}_{GI}. \tag{13}$$

The standard error of the bias-adjusted associations is calculated as shown in line 9. The bias-adjusted estimates and their standard error are then returned for all SNPs (line 11).

Fig. 6 shows a graphical illustration for the Slope-Hunter method using the same data presented in Fig. 5.

**Choice of threshold for inclusion of genetic variants.** Our analysis excluded SNPs that have not achieved a selection threshold in the GWAS of I, i.e. points for which $p$-value $> \lambda$ (our main analysis used $\lambda = 0.001$ as the default threshold). Exclusion of SNPs using a lower (closer to 0) threshold, that includes fewer SNPs, could reduce the efficiency by decreasing the size of the underlying cluster $G_I$ affecting I only. When SNPs are included using a higher (closer to 1) threshold, that includes more SNPs, it is likely that a fraction of these additional SNPs will not affect I (null SNPs). In the presence of these null SNPs, our model (Eq. (12)) is not correct, providing only an approximation of the underlying full model shown in Eq. (10). However if the majority of the null SNPs belong to the cluster ($G_.$) affecting neither I nor P, one would expect an enrichment of the probability concentration $\pi_4$ of SNPs scattered around the origin. Since our model identifies the $G_I$ SNPs as the points scattered closely around the line $\hat{\beta}'_{GP} = b_1 \hat{\beta}_{GI}$, which goes through the origin by definition, such an enrichment is approximately captured by $\pi_1^*$, i.e., $\pi_1^* \approx \pi_1 + \pi_4$ at the true slope ($b_1$). Nevertheless, if a very large $\lambda$ is used, then large value of $\pi_4$ may result in biased or imprecise estimations of $\pi_1$ and $b_1$. One might expect there would be an optimal threshold for SNP selections as is typically observed in risk prediction using polygenic risk scores[21,26]. The effect of modifying this threshold is examined in simulation studies (see 'Simulation setup').

**Underlying assumptions.** Our analytic approach assumes the SNPs are mutually independent of one another, do not interact with the confounders, have linear effects on I and P, and have no interaction with I in their effect on outcome. Our framework assumes a linear effect of I on P. However, the size of that effect is not important for our theoretical developments and it might be zero. Our bias-correction method is robust to violations of the InCLUDE assumption ($A_1$) formulated by Dudbridge et al.[7]. Our procedure assumes constant confounding effect —hence constant correction factor—across all SNPs (Assumption $A_2$). Our model setup implies ZEro Modal Residual Assumption (ZEMRA), which requires that the largest number of similar individual-SNP ratios for SNP-P to SNP-I associations ($\hat{\beta}'_{GP}/\hat{\beta}_{GI}$) comes from the cluster of SNPs only affecting I, even if the majority of SNPs have direct effects on both I and P. This assumption resembles the zero modal pleiotropy assumption (ZEMPA) required by the mode-based estimator for Mendelian randomisation (MR) analyses[27], but with respect to the residual ($e_G = \hat{\beta}'_{GP} - b_1 \hat{\beta}_{GI}$) in our context rather than the pleiotropy. The residual $e_G = 0$ for the first ($G_I$) and fourth ($G_.$) clusters, and equals the true direct effects ($\hat{\beta}_{GP}$)

---

**Box 1. | Slope-Hunter algorithm—adjustment for collider bias in GWAS of conditional analyses**

**Inputs** : $\hat{\beta}_{GI}$ (estimates of SNP-$I$ associations), $\hat{\beta}'_{GP}$ (biased estimates from GWAS of $P$ conditional on $I$), $s_{GI}$ and $s'_{GP}$ (their standard errors) for a set ($G$) of $M$ independent SNPs; a $p$-value threshold $\lambda$.

**Outputs**: bias-adjusted estimates and their standard errors, $\hat{\beta}_{GP}$ and $s_{GP}$.

1 **foreach** $g \in G$ **do**
   // Calculate the $p$-value for the $g - I$ association

2   $p_{gI} = p\left(\chi^2_{(1)} > \left(\frac{\hat{\beta}_{gI}}{s_{gI}}\right)^2\right)$         // $\chi^2_{(1)}$ distributed as Chi-squared with one degree of freedom

3 **end**

4 $G^* = \{g \in G \,|\, p_{gI} < \lambda\}$                  // The set of SNPs retained in our analysis

5 Perform maximum-likelihood using the EM algorithm to fit the model presented in Equation 12 on data from $G^*$, then obtain $\hat{b}_1$, $\hat{\sigma}_{IP}$ and $\hat{\pi}^*_1$.

6 Estimate $s\left(\hat{b}_1\right)$ using the Bootstrap technique         // Standard error of the estimated adjustment factor, $\hat{b}_1$

7 **foreach** $g \in G$ **do**

8   $\hat{\beta}_{gP} = \hat{\beta}'_{gP} - \hat{b}_1\hat{\beta}_{gI}$                  // Calculate the bias-adjusted estimates

9   $s_{gP} = \sqrt{\left(s'_{gP}\right)^2 + \hat{b}_1^2 \cdot s_{gI}^2 + \hat{\beta}_{gI}^2 \cdot s\left(\hat{b}_1\right)^2 + s_{gI}^2 \cdot s\left(\hat{b}_1\right)^2}$     // Estimate standard error of the adjusted associations

10 **end**

11 **return** $\hat{\beta}_{GP}$ and $s_{GP}$

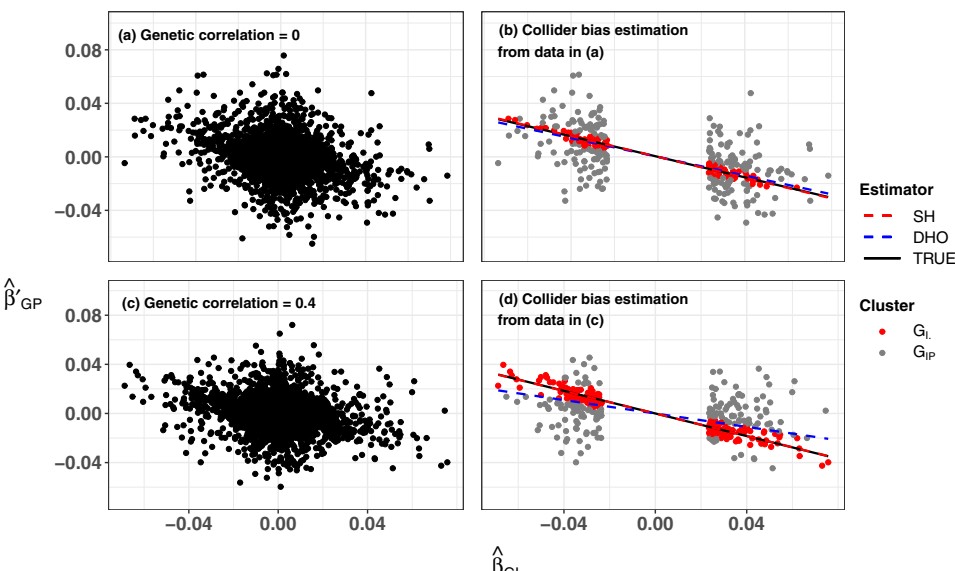

**Fig. 6 A graphical illustration for the Slope-Hunter approach.** Slope-Hunter is applied on estimates of SNP association coefficients with a quantitative trait (I), $\hat{\beta}_{GI}$, and a quantitative outcome ($P$) conditional on I, $\hat{\beta}'_{GP}$ that are simulated for 10,000 independent SNPs from a dataset of 20,000 individuals, with: (a) no genetic correlation between SNP effects on I and P; (c) correlated genetic effects on $I$ and $P$. In both input datasets, depicted in (**a**) and (**c**), the SNP associations are simulated under a hypothesised four-component model for effect-size distribution in which 5% of SNPs have effects on $I$ only, 5% on $P$ only, 5% on both and 85% on neither. The heritability of $I$ and $P$ is 50% and the non-genetic common factors explain 40% of variation in both $I$ and $P$. The fitted correction factors estimated using the Slope-Hunter (SH) method are shown in (**b**) and (**d**) for the input data in (**a**) and (**c**), respectively. After excluding the SNPs that are not associated with $I$, using a $p$-value threshold ($p > 0.001$), the SH method identifies the cluster of variants affecting $I$ only ($G_I$, depicted in red points) and the cluster affecting both $I$ and $P$ ($G_{IP}$, depicted in grey points). The true and estimated correction factors using the SH and the `Hedges-Olkin' estimator of the Dudbridge et al. (DHO)[7] methods are represented by the slopes of the solid black, dashed red and dashed blue lines, respectively. These slopes are $-0.383$, $-0.388$ and $-0.349$ in (**b**) and $-0.460$, $-0.458$ and $-0.273$ in (**d**) for the true, SH and DHO estimators, respectively. Source data are provided as a Source Data file.

which follows the normal distribution $N \sim \left(0, \sigma_P^2\right)$ for the second ($G_{IP}$) and third ($G_P$) clusters (see Eq. (9))[21]. Hence, the most common value of $e_G$ is 0 under our model setup. As an example for when ZEMRA can be violated, suppose a group of SNPs affect $I$ and $P$ via the same exposure, with larger size and larger effect-sizes on $I$ than the $G_I$ cluster of SNPs. Then, it would be expected that a larger number of points, with large enough effect-sizes, lie on a line whose slope $b'$ is different from the true correction factor, i.e. $b' \neq b_1$, leading to violation of the ZEMRA.

**Simulation setup**. We conducted simulation studies to examine the performance of the proposed method under various scenarios. We simulated three main scenarios, each with 10,000 independent SNPs under Hardy-Weinberg equilibrium with minor allele frequencies drawn from a uniform distribution over the interval [0.01, 0.49]. For all scenarios, both $I$ and $P$ were simulated as quantitative traits, the heritability under models shown in Eqs. (1) and (2) was 50%, the non-genetic confounder explained 40% of variation in both $I$ and $P$ with positive effects, and SNP effects, confounders and residual variations ($\varepsilon_I$ and $\varepsilon_P$) were drawn from normal distributions. Under all scenarios, data were simulated for 20, 000 unrelated individuals.

The settings of these scenarios are described in Table 3. In all the main scenarios (1–3), 5% of SNPs (500 SNPs) had effects on $I$ only ($G_I$), 5% on both $I$ and $P$ ($G_{IP}$), 5% on $P$ only, and 85% on neither $I$ nor $P$. The $G_I$ SNPs explain more, equal, and less variation in $I$, compared with the proportion explained by the $G_{IP}$ SNPs in scenarios 1, 2 and 3, respectively. This implies collider bias due to polygenic common effects and non-genetic common factors of $I$ and $P$, that together explain $0.15 + 0.40 = 55\%$, $0.25 + 0.40 = 65\%$, $0.35 + 0.40 = 75\%$ of variation in the outcome under the scenarios 1–3, respectively. The second scenario mimics the simulation study conducted by Dudbridge et al.[7].

We further evaluated the performance of the proposed method when there are fewer SNPs affecting $I$ only (1%), compared with the number of SNPs affecting both traits $I$ and $P$ (9%) using two scenarios: scenario S1; scenario S2, with different and equal proportions, respectively, of the variation in $I$ explained by the $G_I$ and $G_{IP}$ clusters (see Supplementary Table 11).

In all main and secondary scenarios, effects of the $G_{IP}$ SNPs on $I$ were simulated independently from their effects on the outcome $P$ (i.e. genetic correlation of the direct effects on $I$ and $P$ ($\rho_d$) is zero) satisfying the InCLUDE assumption ($A_1$). All simulations were repeated with correlated effects on $I$ and $P$, whereby effects of the $G_{IP}$ SNPs were drawn from a bivariate normal distribution with a correlation coefficient $\rho_d = 0.9, 0.5, -0.5, -0.9$. This violates the InCLUDE assumption providing correlated direct effects with a correlation direction as the same as (for positive $\rho_d$ values) and opposite to (for negative $\rho_d$ values) the correlation ($\rho_c$) due to the induced association with $P$ as a result of collider bias.

Additionally, we simulated further two secondary scenarios: scenario S3; scenario S4 (the latter violates the ZEMRA assumption), where the effects of the $G_{IP}$ SNPs on $I$ and $P$ act via a common exposure explaining lower (scenario S3) and larger (scenario S4) variation in $I$ than that explained by the $G_I$ SNPs, and inducing perfectly correlated direct effects with same or opposite correlation direction to the direction of $\rho_c$, Supplementary Table 11.

Estimated SNP effects on $I$, $\hat{\beta}_{GI}$, were obtained from linear regression of $I$ on genotype, whereas the estimates $\hat{\beta}'_{GP}$ were obtained from linear regression of $P$ on genotype conditional on $I$. For each scenario, we performed 1000 simulations and reported the mean of the 1000 within-simulation differences between estimated correction factors and the true collider bias. The type-1 error rates of SNP associations with $P$ were evaluated at $p < 0.05$. Since the collider bias is proportional to the effect on $I$, see Eq. (9), type-1 error rates vary among SNPs with different effects on $I$.

Therefore, we estimated: the mean type-1 error over all SNPs with no effect on $P$ (i.e. the clusters $G_I$ and $G_.$); the mean type-1 error over SNPs with effects on $I$ only (i.e. the $G_I$ cluster) because the $G_.$ SNPs do not suffer bias and they can dominate $G_I$ SNPs, when combined, due to cluster sizes. We estimated the family-wise type-1 error over the $G_I$ cluster, as the proportion of simulations in which at least one variant had $p < 0.05$ after Bonferroni multiple-testing correction for the number of SNPs. The mean power over all SNPs with effects on $P$ (the clusters $G_{IP}$ and $G_P$) and over SNPs with effects on both $I$ and $P$ ($G_{IP}$) were estimated. The mean absolute bias and mean square error across all SNPs, and across SNPs with effects on $I$ were estimated.

Results from the Slope-Hunter method ('SH' estimator) were compared with the unadjusted estimator and the estimator of the method of Dudbridge et al.[7] with Hedges-Olkin adjustment (DHO estimator) and with simulation extrapolation adjustment (DSIMEX estimator) for regression dilution[7]. Because DHO and DSIMEX results were almost identical, we only reported the DHO results. Furthermore, the individual SNP with highest type-1 error for the unadjusted estimator was identified and compared with the type-1 error of the adjusted estimators. We identified SNPs with the greatest increase and decrease in power between the unadjusted estimator and all estimators of the adjusted analyses using SH and DHO. The mean of maximum absolute bias was also compared between the unadjusted and adjusted estimators.

We explored the capability of our method to handle situations with different threshold values ($\lambda$) by varying the $p$-value threshold in the $z$-test for SNP selection from the GWAS of $I$. We used $\lambda = 10^{-5}, 10^{-4}, 10^{-3}, 10^{-2}$, and $10^{-1}$ under Scenario 2. Then, we studied the bias in correction factors and the type-1 error

**Table 3 Descriptions of the main simulated scenarios in which true effect-sizes were simulated for 10, 000 independent SNPs under the hypothesised four-component mixture model (Eqs. (8a)–(8d)).**

| Sc. | Cluster sizes | | | | Explained I's variation | | Genetic correlation | | True slope |
|---|---|---|---|---|---|---|---|---|---|
| | $\pi_{G_I}$ | $\pi_{G_{IP}}$ | $\pi_{G_P}$ | $\pi_{G_.}$ | $R^2_{G_I}$ | $R^2_{G_{IP}}$ | Direction | Coefficient ($\rho_d$) | |
| 1 | 0.05 | 0.05 | 0.05 | 0.85 | 0.35 | 0.15 | $\rho_d$ & $\rho_c$ are in **the same** direction | 0.9 | −0.48 |
| | | | | | | | | 0.5 | −0.43 |
| | | | | | | | Uncorrelated direct effects | 0 | −0.37 |
| | | | | | | | $\rho_d$ & $\rho_c$ are in **opposite** directions | −0.5 | −0.30 |
| | | | | | | | | −0.9 | −0.25 |
| 2 | 0.05 | 0.05 | 0.05 | 0.85 | 0.25 | 0.25 | $\rho_d$ & $\rho_c$ are in **the same** direction | 0.9 | −0.55 |
| | | | | | | | | 0.5 | −0.47 |
| | | | | | | | Uncorrelated direct effects | 0 | −0.37 |
| | | | | | | | $\rho_d$ & $\rho_c$ are in **opposite** directions | −0.5 | −0.26 |
| | | | | | | | | −0.9 | −0.17 |
| 3 | 0.05 | 0.05 | 0.05 | 0.85 | 0.15 | 0.35 | $\rho_d$ & $\rho_c$ are in **the same** direction | 0.9 | −0.62 |
| | | | | | | | | 0.5 | −0.51 |
| | | | | | | | Uncorrelated direct effects | 0 | −0.37 |
| | | | | | | | $\rho_d$ & $\rho_c$ are in **opposite** directions | −0.5 | −0.22 |
| | | | | | | | | −0.9 | −0.08 |

Sc. = scenario; $\pi_{G_I}$, $\pi_{G_{IP}}$, $\pi_{G_P}$, $\pi_{G_.}$ = proportion of SNPs affecting $I$ only, both $I$ and $P$, $P$ only, and neither $I$ nor $P$, respectively; $R^2_{G_I}$, $R^2_{G_{IP}}$ = proportion of variation in $I$ explained by the clusters $G_I$ and $G_{IP}$, respectively; $\rho_d$ = genetic correlation due to the direct effect on $I$ and $P$; $\rho_c$ = genetic correlation due to the induced association with $P$ as a result of the collider bias effects; True slope = true correction factor required for adjusting the collider bias. The SNPs were simulated under Hardy-Weinberg equilibrium with minor allele frequencies drawn from a uniform distribution over the interval [0.01, 0.49]. Both $I$ and $P$ were simulated as quantitative traits with heritability of 50%. The non-genetic confounder explained 40% of variation in both $I$ and $P$. The SNP effects, confounders and residual variation, $\varepsilon_I$ and $\varepsilon_P$, were drawn from normal distributions. Data were simulated for 20, 000 unrelated individuals. In each scenario, a collider bias is developed due to confounders that require a certain correction factor—shown in the last column—to adjust for the induced collider bias.

rates for resulting Slope-Hunter estimates compared with estimates from the unadjusted analyses and alternative bias-correction methods. Since the winner's curse problem may produce bias when selection of variants and estimation of their associations are performed based on the same study, we further examined the performance of the Slope-Hunter method when the effect-sizes of the SNP associations with $I$ are estimated using an independent dataset from the one used to select the SNPs[21].

**Genetic factors causing fasting insulin independently of body mass index**. We applied Slope-Hunter to publicly available GWAS summary-level data for fasting blood insulin (FI) level conditional on body mass index (BMI) using $\lambda = 0.001$. The aim of this analysis is to identify genetic effects on FI that do not act through BMI. We downloaded summary statistics for GWAS studies of BMI[16] and BMI-adjusted FI[1], harmonised the data, and analysed 21,779 variants present in both datasets. We created an LD-pruned set of SNPs with $R^2$ threshold of 0.1 within 250 SNP windows. This set contained 12,792 SNPs that were then considered for estimating the collider bias adjustment factor. The LD-pruning was estimated using the European ancestry population of the 1000 Genomes reference[28], which has similar ancestry to the BMI and conditional analysis of FI GWAS. The pruning was performed on random basis, rather than based on p-values, to avoid the winners curse bias problem[29]. The results obtained from the Slope-Hunter method were compared with results from the naive conditional analysis and DHO method.

**Case-only study of breast cancer mortality**. We downloaded publicly available GWAS summary statistics for breast cancer incidence[17] (14,910 cases and 17,588 controls) and mortality[5] and considered 13,783,685 variants present in both datasets. After harmonising the case/control and mortality data, we analysed 10,202,280 variants. An LD-pruned set of SNPs, contained 94,744 SNPs, was formed using $R^2$ threshold of 0.1 within 250 SNP windows. The LD-pruning was estimated using the European ancestry population of the 1000 Genomes reference[28], and was performed based on a random selection for the pruned in SNPs. The results obtained from the Slope-Hunter method using $\lambda = 0.001$ were compared with results from the naive conditional analysis and DHO method.

**Reporting summary**. Further information on research design is available in the Nature Research Reporting Summary linked to this article.

## Data availability
The BMI data that support the findings of this study are available from "https://portals.broadinstitute.org/collaboration/giant/index.php/GIANT_consortium_data_files". The BMI-adjusted fasting blood insulin data are available from "https://www.ebi.ac.uk/gwas/publications/25625282". The summary-level data of breast cancer GWAS and of breast cancer mortality are available from "http://bcac.ccge.medschl.cam.ac.uk/bcacdata/oncoarray/oncoarray-and-combined-summary-result". Source data are provided with this paper.

## Code availability
All procedures described in this manuscript have been implemented into an open source R package named 'SlopeHunter'[23] that is available from https://github.com/Osmahmoud/SlopeHunter. All analyses have been conducted using R 4.0.4 that is available from https://cran.r-project.org/bin/windows/base/old/4.0.4/.

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

## Acknowledgements
F.D. was supported by the Medical Research Council (MRC) [grant number: MR/S037055/1]. G.D.S., M.M. and K.T. work in the Medical Research Council Integrative Epidemiology Unit at the University of Bristol [MC_UU_00011/1], [MC_UU_00011/7] and [MC_UU_00011/3], respectively.

## Author contributions
O.M. and K.T. conceived and designed the study. O.M. planned, designed and developed the proposed method 'Slope-Hunter', wrote the software, performed the simulation studies, carried out the statistical data analyses, and drafted the manuscript. F.D. and K.T.

contributed to the study design and result presentations. K.T. supervised the statistical analyses. O.M., F.D., G.D.S., M.M. and K.T. contributed to the interpretation of the results, critically reviewed the manuscript, and approved the final version.

## Competing interests

The authors declare no competing interests.
