## [Peer Review File · Nature Communications]

Reviewers' comments:

Reviewer #1 (Remarks to the Author):

The method introduced in this paper addresses an interesting and relevant problem, estimating variant association with prognosis conditional on incidence. This problem is challenging because, if confounders, U affecting both prognosis and incidence exist, then conditioning on incidence introduces collider bias which can lead to significant errors in effect estimation. The problem has previously been addressed by Dudbridge et al, however, the authors show that that solution is sensitive to violations of strong and plausibly unrealistic assumptions. The main idea of this proposal is to use variants that affect incidence only to estimate the event bias and then use this estimate to correct effect estimates for all variants. This is a sound idea and seems to perform reasonably well in simulations. However, in several places the exposition is muddled, and it is not always clear what assumptions are relied on. The simulations are somewhat unrealistic, and the results presentation is hard to follow. With improvements to these, I think the work could be a very useful contribution to the field.

Major comments

1. I think there is some difference between the model in (1) and (2) and the example of binary incidence and prognosis that is used throughout the paper. In this example, there is a binary disease, I , and a prognosis, P , that (in many cases) can only occur if the disease has already occurred. This implies that there should be an interaction between G and I in (2) since, if P is always 0 if I is 0, G can only affect P if I is 1. It's be easy to imagine such an interaction even if the probability of $P | I = 0$ is not 0 or if I is continuous. For example, heart disease risk variants may have different effects among diabetic and non-diabetic individuals.

One important point in the binary trait case is that the target of inference isn't really β_{GP} , but β_{GP} given $I = 1$ (i.e. the genetic effect on prognosis among affected individuals). I think it would be useful to discuss this and add a statement saying that (2) is assumed for some relevant range of I and the target of inference is the effect of G on P given that I is in this range (which could be the entire range if I is not restricted).

2. I find the discussion of violation of A2 confusing and in places contradictory:

a. Page 5 "Whilst for a single SNP $G2$ affecting both incidence and prognosis (Figure 2 and Figure 3), the genetic component of U equals entire shared genetic basis of I and P , minus the component attributed to the SNP under consideration, $G2$." Is confusing.

In Figures 2 and 3, U does not have any genetic component in that there are no variants that affect U . There is correlation between $G2$ and U induced by conditioning on I .

This statement implies that there will be heterogeneity in b across all variants affecting both I and P which is at odds with a later statement

"When SNP effects on incidence and prognosis are uncorrelated, the index event bias should be exactly the same for various set sizes of $G1$ compared with $G2$ as the confounding effect would be entirely due to the non-genetic component that is equally simulated across all scenarios."

This second statement is a bit hard to parse but seems to imply that a condition for heterogeneity in b is correlation between β_{GI} and β_{GP} .

To me it seems that differences in b across variants occur in the scenario in Figure 3 but not in Figure 2 as long as, in Fig. 2, the effects of $G2$ on I and of $G2$ on P are independent.

b. Slope-Hunter doesn't appear to be robust to violation of A2. In (8) all variants in $G2$ are assumed to have the same value, b_2 and then it is further assumed that $b_1 \approx b_2$. If this is an assumption of the method it should be included in the Underlying Assumptions section.

3. (8) implies a four component mixture model. It seems odd to first fit it with a two component mixture ($G4$ and not $G4$) in that I would expect the non- $G4$ cluster to not look normally distributed. I am curious how this affects the accuracy of the $G4$ classification. Relatedly, $G3$ is identified simply based on a p-value threshold. Why can't $G4$ be identified the same way? An alternative that is more unified and flexible would be to estimate an empirical effect size prior (for

example a mixture of many bivariate normals) and then estimate posterior probabilities of belonging to G4 and G3.

4. Figure 6d. The class labeled G1 contains many SNPs with $\hat{\beta}_{GI}$ close to zero while in figure 6c all such variants are assigned to G4. I think I understand that these are the lambda fraction of G4 retained but it is confusing to now refer to them as G1. Are the black points in 6d actually G*?

5. What is the rationale for the parameters in the simulation scenarios? In particular, In Sc.1 and Sc.3 there are very few variants. This will lead to much larger I effect sizes for G1 variants than for G2. This seems unrealistic to me as I might expect G1 and G2 variants to have similar I effect sizes. I would also imagine that this could make G1 easier to identify. It would be good to explain the effect of varying each parameter (number of variants, total heritability etc.). If possible, if some of the parameters can be held across scenarios it would be easier to understand the effect on relative performance of each variable.

6. Page 11. "In this case, if the class G2 explains more of the variation in incidence than the class G1, the Slope-Hunter may be severely biased because it would completely swap the classes rather than having affordable misclassification error as". My understanding (and also what is stated in the discussion) is that G2 is a group of SNPs that cluster around the slope b_1 line but have higher variance. Since G2 has higher variance in β_{GP} than G1, how can the two classes be swapped? Could the expectation of a higher variance in G2 be used to identify instances when the classes are swapped?

7. In Table 3, I assumed that type 1 error and power referred to a test of whether $\beta_{GP} = 0$. It seems weird to give T1E for G1 + G2 because only G1 has $\beta_{GP} = 0$. Similarly it seems odd to give power for the two groups combined since only G2 has $\beta_{GP} \neq 0$.

8. I don't understand the suggestion in the discussion of using the technique for MR. There are definitely thematic connections – use of mixture models and hidden confounders and I like that these are pointed out. But it seems to me the problem here is fundamentally different. If there is a direct application, it would be good to spell that out more clearly.

Minor Comments

9. A question about presentation of simulation results. I remain a bit confused about when b differs across variants and when it does not. It seems that it would when β_{GP} and β_{GI} are correlated or in Sc.6. In these cases, is the average difference between estimated and true event bias averaged over snps as well as over simulations? Some clarification on this issue would help.

10. Page 8. Dudbridge et al argue that the Hedges-Olkin estimator has too large variance and use SIMEX instead. A comment about why the H-O estimator works here could be useful.

11. It would be nice if the information in the tables could be displayed graphically.

12. Discussion "whereby independent causes of the incidence become correlated when selecting only on cases and then may confound analysis of prognosis." Is hard to understand.

13. It could be useful in motivating the problem and highlighting its significance to point out that this problem can occur essentially whenever a GWAS is conducted conditioning on a heritable trait and not all of these would typically be considered "prognosis". For example, the problem could occur looking at waist to hip ratio conditional on BMI.

Signed,
Jean Morrison

Reviewer #2 (Remarks to the Author):

Mahmoud et al present a new method to correct association summary statistics of subsequent traits. The methodological foundation of the problem is solid and the proposed method seems logical. However, I found several weaknesses of this paper: (i) the paper difficult to read; (ii) the propose algorithm being quite ad hoc with many arbitrary choices (for strategies and parameters); (iii) the simulations are not presented clearly and the reader is left without a detailed constellation of parameters when and why would the SH method break down. Moreover, presenting no application to real data is a substantial weakness of the work and makes it much less appealing to a journal like Nature Communications, which has very broad readership. Below I proved a detailed list of questions and suggestions to improve the work and its presentation.

Major comments

1. It is not clear how pruning is applied (if P-values are used to determine the order, which ones G-P or G-I) and whether it would introduce bias (winners curse).
2. While the general strategy is clear and logical, but the actual algorithm needs to be explained much clearer. Here are a few points that in my opinion need more clarification:
 - a. "Exclusion of all SNPs assigned to the class G4 may discontinue the dimensional space" - I'm lost what it means.
 - b. "fitting a clustering solution fj" - What is a clustering solution?
 - c. What is the rationale behind the definition of ω_g ? Why is the second term not squared?
 - d. The notations G1, G2, G3, G4 are not intuitive, please consider renaming these groups (e.g. G_I for incidence SNPs, G_IP for SNPs associated with both, etc.)
 - e. There are arbitrary parameters (δ , ϵ), how robust it the method to variations in these values?
 - f. In step 4 (bottom of page 7), it is not clear how the cluster centres are calculated (effect directions are arbitrary, aren't mean effects expected to be zero?)
 - g. What is the rational for step 4, why situations where clusters G1, G2 that are far from each other are not considered?
 - h. The pseudo-code is much clearer than the text description
3. The algorithm is not intuitive, why not fitting all (pruned) SNPs together at the same time using a 4 component bivariate Gaussian mixture model?
4. I major weakness of the paper that is presents no application to real data, which would be extremely crucial to convince the readers that the formulated problem is important and the implications of the proposed correction are far-reaching.
5. An application would be particularly useful because one could see the impact of effect size bias on downstream analysis, such as MR.
6. Tables 3-7 are very boring presentations of the results. Most of these should be move to the supplement and the authors should try to present some of these comparisons more succinctly as figures.
7. Figures 1-3 & 5 should be merged. I don't find any reference to Fig 8/9 in the text and do not find the plots very meaningful either. Either move it to supplement (and refer to it) or explain them much better and motivate their presence in the main text. I don't see much point of Figure 7 either, it does not provide too much insight into the hypothetical covariance structure of the effect estimates. Could it be coupled with Fig 4? Not sure why it should be a main figure. Figure 6 is great and very didactic.
8. In the different scenarios (1-5), it is not clear which is the crucial parameter that leads to deteriorating performance of the SH method? G1 explaining larger variance is clearly an advantage, but what role the polygenicity of incidence play here? Also, does the relative heritability of the G1 and G2 groups important? The authors provide little intuition in the Discussion.
9. I don't understand how the SH method can maintain good control of the type I error even in scenario 5 and high genetic correlation, despite providing a poor estimate for the adjustment factor?

Minor comments

1. The cluster parameters are mentioned strangely (parameters look like if they were a product) in the sentence starting "The geometric features of each cluster are determined by its covariance matrix that can be decomposed to the form of...". Also it has θ_k twice.
2. Could the authors elaborate why their method is robust against sample overlap
3. "assuming that $b_1 \sim b_2$, which is typically the case for small effects as in GWAS" – Could the authors explain why?
4. Can the authors explain, why it was not necessary to vary the number of markers in the G3 group (and their explained variance)?

Point-by-point response to referees

NCOMMS-20-04869 - Slope-Hunter: A robust method for index-event bias correction in genome-wide association studies of conditional outcomes

We are grateful to the reviewers, the senior editor (SE) and to the editor for the detailed and helpful comments and suggestions. In the following, our responses are in bold blue letters and the comments of the reviewers in black italic letters.

Reviewer 1:

The method introduced in this paper addresses an interesting and relevant problem, estimating variant association with prognosis conditional on incidence. This problem is challenging because, if confounders, U affecting both prognosis and incidence exist, then conditioning on incidence introduces collider bias which can lead to significant errors in effect estimation. The problem has previously been addressed by Dudbridge et al, however, the authors show that that solution is sensitive to violations of strong and plausibly unrealistic assumptions. The main idea of this proposal is to use variants that affect incidence only to estimate the event bias and then use this estimate to correct effect estimates for all variants. This is a sound idea and seems to perform reasonably well in simulations. However, in several places the exposition is muddled, and it is not always clear what assumptions are relied on. The simulations are somewhat unrealistic, and the results presentation is hard to follow. With improvements to these, I think the work could be a very useful contribution to the field.

Major comments:

(C1) I think there is some difference between the model in (1) and (2) and the example of binary incidence and prognosis that is used throughout the paper. In this example, there is a binary disease, I , and a prognosis, P , that (in many cases) can only occur if the disease has already occurred. This implies that there should be an interaction between G and I in (2) since, if P is always 0 if I is 0, G can only affect P if I is 1. It's be easy to imagine such an interaction even if the probability of $P | I = 0$ is not 0 or if I is continuous. For example, heart disease risk variants may have different effects among diabetic and non-diabetic individuals.

One important point in the binary trait case is that the target of inference isn't really β_{GP} , but β_{GP} given $I = 1$ (i.e. the genetic effect on prognosis among affected individuals). I think it would be useful to discuss this and add a statement saying that (2) is assumed for some relevant range of I and the target of inference is the effect of G on P given that I is in this range (which could be the entire range if I is not restricted).

(R1) We described the problem more generally, and as pointed out in this comment, revised the example of binary incidence and prognosis that was used throughout the manuscript. As suggested by the Reviewer 1 (comment C13) we redefined the underlying problem by amending the 'Background' section (page 2) to read:

"There is increasing interest in the use of genome wide association studies (GWAS) conditioned on a phenotype, such as a GWAS of blood pressure conditional on body mass index (BMI) so as to avoid only genetic variants associated with BMI appearing important. An example of such conditional analyses that is increasingly being used is GWAS of prognosis ... Such an analysis is referred to as 'conditional analysis' throughout this manuscript"

We have accordingly expressed the models of an outcome (P) with a continuous incidence trait (Equation 2) and binary incidence trait (Equation 3) including explicit interpretation for the target inference in each case (page 4):

“Likewise if P is binary ... The effect of our interest is the direct effect of the SNP on outcome conditional on I and confounders U , i.e. β_{GP} in 2 and 3. However in practice, we can only estimate the SNP-outcome association conditional on I , as all relevant confounders may not be observed: ...”

We have explicitly stated that these models assume no interaction between G and I as mentioned in the ‘Underlying assumption’ (page 11):

“Our analytic approach assumes the analysed SNPs are mutually independent of one another, do not interact with the confounders, have linear effects on incidence and outcome, and have no interaction with incidence in their effect on outcome.”

(C2) I find the discussion of violation of A2 confusing and in places contradictory:

a. Page 5 "Whilst for a single SNP G_2 affecting both incidence and prognosis (Figure 2 and Figure 3), the genetic component of U equals entire shared genetic basis of I and P , minus the component attributed to the SNP under consideration, G_2 ." Is confusing. In Figures 2 and 3, U does not have any genetic component in that there are no variants that affect U . There is correlation between G_2 and U induced by conditioning on I .

This statement implies that there will be heterogeneity in b across all variants affecting both I and P which is at odds with a later statement

"When SNP effects on incidence and prognosis are uncorrelated, the index event bias should be exactly the same for various set sizes of G_1 compared with G_2 as the confounding effect would be entirely due to the non-genetic component that is equally simulated across all scenarios."

This second statement is a bit hard to parse but seems to imply that a condition for heterogeneity in b is correlation between β_{GI} and β_{GP} . To me it seems that differences in b across variants occur in the scenario in Figure 3 but not in Figure 2 as long as, in Fig. 2, the effects of G_2 on I and of G_2 on P are independent.

b. Slope-Hunter doesn't appear to be robust to violation of A2. In (8) all variants in G_2 are assumed to have the same value, b_2 and then it is further assumed that $b_1 \approx b_2$. If this is an assumption of the method it should be included in the Underlying Assumptions section.

(R2)

- a. The definition of U implies that it is a composite variable of all common causes of I and P including the common polygenic effect (i.e. the genetic variants with effect on both I and P). We have made that clearer by adding a statement in the ‘Methods’ (page 4):

“... it is assumed that a continuous trait I is a linear function of the coded genotype G , U (common causes of trait I and outcome, including polygenic effects and non-genetic factors)...”

Moreover, we amended the caption of Figure 1 to read:

“... U is a composite variable including all common causes of I and P , involving common polygenic effect on I and P as well as non-genetic factors ...”

We have amended the 'Underlying assumptions' (page 11) to read:

"For a single SNP G_i affecting incidence only, as in Figure 1(a), the genetic component of U is the entire shared genetic basis of I and P (i.e. common polygenic effect). Therefore, the genetic component of U is constant across all SNPs affecting incidence only (i.e. the class G_i). Since the non-genetic component of U is constant across all SNPs by definition, our estimate of the adjustment factor, b_1 , is robust to violations of assumption A2. However, the procedure of correcting bias for all variants assumes a constant confounder effect, U , across all variants (i.e. $b_1 = b_2$ in equation 8 which requires assumption A2 to hold ...".

Furthermore, we clarified the statement quoted by the reviewer at the same paragraph (page 11) to read:

"For a single SNP G_{IP} affecting both incidence and outcome, as in Figures 1(b) and 1(c), the SNP G_{IP} is itself a part of the common polygenic effect on I and P , i.e. contributing to the genetic component of U , unless it is the SNP under consideration. Therefore when considering a single SNP G_{IP} , the genetic component of U is the residual common polygenic effect on I and P that equals the entire shared genetic basis of I and P , minus the component attributed to the SNP under consideration ..."

- b. Since the Slope-Hunter uses just the cluster of variants affecting only incidence, for which U is constant, for estimating the adjustment factor (b_1), then it does not require either assumption A1 or assumption A2 to obtain an unbiased estimate of the index event bias. However, assumption A2 is necessary for using the Slope-Hunter's estimated adjustment factor to correctly obtain bias-adjusted estimates for the variants affecting both incidence and prognosis. Therefore, we have added a statement to the 'Underlying assumptions' (page 11) explaining that:

"... For a single SNP G_I affecting incidence only, as in Figure 1(a), the genetic component of U is the entire shared genetic basis of I and P (i.e. common polygenic effect). Therefore, the genetic component of U is constant across all SNPs affecting incidence only (i.e. the class G_I). Since the non-genetic component of U is constant across all SNPs by definition, our estimate of the adjustment factor, b_1 , is robust to violations of assumption A2. However, the procedure of correcting bias for all variants assumes a constant confounder effect, U , across all variants (i.e. $b_1 = b_2$ in equation 8 which requires assumption A2 to hold ...".

(C3) (8) implies a four component mixture model. It seems odd to first fit it with a two component mixture (G_4 and not G_4) in that I would expect the non- G_4 cluster to not look normally distributed. I am curious how this affects the accuracy of the G_4 classification. Relatedly, G_3 is identified simply based on a p-value threshold. Why can't G_4 be identified the same way? An alternative that is more unified and flexible would be to estimate an empirical effect size prior (for example a mixture of many bivariate normals) and then estimate posterior probabilities of belonging to G_4 and G_3 .

(R3) Equation (8) does not represent our model of interest. It rather presents the mathematical forms of the conditional estimate defined by Dudbridge et. al. (2019) – shown in Equation (5) – corresponding to different statuses of variants in terms of their effects on I and P . The main purpose of the Equation (8) is to illustrate how consideration of only the class G_i - in estimating the adjustment factor - would

be robust against the assumptions A1 (InCLUDE), that is required by the method of Dudbridge et. al. (2019).

Our aim was to identify the pattern of the class G_1 (renamed to G_I in the revised version) rather than identifying memberships of the four classes. Therefore, the accuracy of the G_4 (renamed $G_{..}$) and G_3 (renamed G_P) classifications were out of our scope because they do not suffer the index event bias. As a procedure for pre-processing data prior to our main analysis, we firstly fit a two-component mixture model ($G_{..}$ and $\hat{G}_{..}$) to exclude data points that scattered around the origin in a noisy form. We revised our text in page 7 to clarify that:

“... corresponding points plotted on the Cartesian coordinate system are expected to scatter around the origin in a noisy form with a high probability concentration since the majority of variants in GWAS studies are expected to belong to this cluster, Figure 3(b).”

The class ‘ $G_{..}$ ’ has not been identified based on p-value thresholds, as for the class ‘ G_P ’, because there would be two thresholds required for this class resulting in an elimination of a rectangle (a square if both thresholds are equal) of data points from the dimensional space, around the origin. This could distort the subsequent analysis and influence the identification of target class ‘ G_I ’. Our simulated example, plotted in Figure (3), shows that our pre-processing procedure works well enough in identifying and removing that pattern from the data prior to our main analysis. However, a random subsample of these data points is retained to facilitate pattern recognition of the target class (G_I). The caption of Figure (3) is revised to read as follows:

“... (c) a random sub-sample of $G_{..}$ SNPs is retained in the analysis, whereas the remaining variants in this cluster and the variants G_P affecting P only are excluded. The latter is identified using a P-value threshold for SNP-incidence associations; (d) The class of variants affecting incidence only, G_I , is identified and an estimate of its linear regression slope ... is obtained to correct for the index event bias of all SNPs ...”

We agree that the non- G_4 (renamed as ‘non- $G_{..}$ ’ in the revised version) cluster could not be normally distributed, but that the assumption of normality of the $G_{..}$ cluster should be adequate for implementing our pre-processing procedure.

(C4) Figure 6d. The class labeled G_1 contains many SNPs with $\hat{\beta}_{G_I}$ close to zero while in figure 6c all such variants are assigned to G_4 . I think I understand that these are the lambda fraction of G_4 retained but it is confusing to now refer to them as G_1 . Are the black points in 6d actually G^* ?

(R4) In the final analysis depicted in Figure 6(d), renumbered as Figure 3(c) in the revised version, the remaining variants are assigned to either the cluster G_1 (renamed G_I in the revised version) or the cluster G_2 (renamed G_{IP} in the revised version). This applies also to the lambda fraction of G_4 (renamed $G_{..}$ in the revised version), represented by the red points in Figure 3(c)), that is retained in the analysis after the pre-processing procedure illustrated in Figures 3(a)-3(c). This lambda fraction of $G_{..}$ could contain variants that affecting only incidence, but with small effects (i.e. members of the true class G_I). These variants are more likely to be assigned to $G_{..}$ in the first stage of our analysis. Therefore, we believe that it is sensible to name variants in the final analysis as clusters G_I and G_{IP} to refer to the target pattern (black points in Figure 3(d)) and the pattern of variants affecting both I and P respectively. Figure 3(d) shows that the estimated slope of our identified target pattern (slope of the dashed green line) could capture the true index event bias (represented by the black sloid line) in the presence of

genetic correlation, where the other method fails. We revised the text explaining these procedures in page 7 to read:

“Unlike $G_{..}$, the variants of the cluster G_i are expected to form a linear pattern whose slope is the adjustment factor to correct the index event bias, see b_1 in equation 8. Our proposal aims to estimate this slope by identifying such a linear pattern. However, excluding all SNPs assigned to the cluster $G_{..}$ may distort such a pattern, e.g. by splitting it into two disconnected ellipses, leading to poor or invalid identification of the cluster G_i To avoid distortion of the G_i pattern, which goes through the origin by definition (see equation 8), our method retains a fraction λ of the SNPs assigned to $G_{..}$.”

and in page 8 to read:

“We define a set ... For each given proportion $\lambda_j \in \Lambda$, the following steps are performed:

1. A sub-sample of SNPs ... of size λ_j is randomly selected from the class $G_{..}$ using the vector of weights ...
2. SNPs with no effect on I among the cluster $\prime\{G_{..}\}$, the complementary cluster of $G_{..}$, are identified and excluded ...
3. ... We fit a cluster-based model, f_j , using the expectation–maximization (EM) algorithm [15] to identify the underlying distributions and to estimate the probability of each SNP belonging to the G_i class ...”

Pattern identification of the target cluster could be still achieved with a reasonable proportion of misclassified SNPs. We have amended the ‘Discussion’ (page 18) to read:

“However, performance of the Slope-Hunter method depends mainly on pattern identification of the G_i class, and hence its slope, and not directly on the classification accuracy of the true G_i variants. Therefore, the Slope-Hunter method can afford a reasonable amount of misclassification as long as the misclassified variants do not severely distort the pattern estimated for the target class ‘ G_i ’. Further studies may be needed to gain more understanding about the influences of misclassification error rates on the bias-adjusted estimates of the association with the outcome”.

We have showed the uncertainty of the variants assigned to the cluster G_i and highlighted the misclassified SNPs in Figure S1 and amended the ‘Results’ (page 15) to read:

“Figure S1 shows the uncertainty about the assignment of variants to the cluster G_i , and highlights the misclassified variants to that cluster, under Sc.2 and genetic correlation of 0.5 at four random simulations out of the 1000 simulations. The estimated adjustment factors were close to the true index event bias, even with many misclassified variants, suggesting that the pattern of the target class could be identified using our approach”.

(C5) What is the rationale for the parameters in the simulation scenarios? In particular, In Sc.1 and Sc.3 there are very few variants. This will lead to much larger I effect sizes for G_1 variants than for G_2 . This seems unrealistic to me as I might expect G_1 and G_2 variants to have similar I effect sizes. I would also imagine that this could make G_1 easier to identify. It would be good to explain the effect of varying each parameter (number of variants, total heritability etc.). If possible, if some of the parameters can be held across scenarios it would be easier to understand the effect on relative performance of each variable.

(R5) As suggested by the reviewer, we have revised the simulated scenarios and added two more scenarios (named Sc.1 and Sc.3 in the revised version). Across Sc.1 – Sc.3, we have kept equal number of variants for G_1 and G_2 (renamed as G_I and G_{IP} respectively) with different explained variation in incidence by each group, see Table 1. We simulated another scenario (Sc.4) in which we increased both size and proportion of explained variation in I for the class G_{IP} violating the main underlying assumption of our approach (referred to as MAPCA). The scenarios in the initially submitted manuscript that were named as Sc.1, Sc.3 and Sc.5 have been renamed and moved to the supplementary materials as Sc.A1, Sc.A2 and Sc.A3 in which the class G_I has fewer SNPs, see Table S1.

(C6) Page 11. “In this case, if the class G_2 explains more of the variation in incidence than the class G_1 , the Slope-Hunter may be severely biased because it would completely swap the classes rather than having affordable misclassification error as”. My understanding (and also what is stated in the discussion) is that G_2 is a group of SNPs that cluster around the slope b_1 line but have higher variance. Since G_2 has higher variance in β_{GP} than G_1 , how can the two classes be swapped? Could the expectation of a higher variance in G_2 be used to identify instances when the classes are swapped?

(R6) We have amended the explanation, provided mathematical representation, and formulated this point into an assumption named the ‘Maximal Predictive Capability Assumption’ (MAPCA) in the ‘Underlying assumptions’ (page 11):

“From equation 8 ... By definition, variants that affect I and P via the same exposure, say E as depicted in Figure 1(c), should have the same value for Δ_g . This leads to an exact proportional relationship between their conditional estimates and their SNP-incidence association, typically like the key characteristic of the G_I class. The ability of our approach to unbiasedly estimate the true index event bias relies on the following fundamental assumption termed the Maximal Predictive Capability Assumption (MAPCA): the class G_I explains more variation in incidence than any subset of the class G_{IP} with the same Δ_g value.”

We have showed the results when this assumption is violated under Sc.4 whereby the cluster G_2 (renamed as G_{IP} in the revised version) might have lower variance than the cluster G_1 (renamed as G_I in the revised version), i.e. the characteristics of both clusters are swapped if this assumption does not hold.

(C7) In Table 3, I assumed that type 1 error and power referred to a test of whether $\beta_{GP} = 0$. It seems weird to give T1E for $G_1 + G_2$ because only G_1 has $\beta_{GP} = 0$. Similarly it seems odd to give power for the two groups combined since only G_2 has $\beta_{GP} \neq 0$.

(R7) We agree with the reviewer and these have been revised for all tables of type 1 error and power.

(C8) I don’t understand the suggestion in the discussion of using the technique for MR. There are definitely thematic connections – use of mixture models and hidden confounders and I like that these are pointed out. But it seems to me the problem here is fundamentally different. If there is a direct application, it would be good to spell that out more clearly.

(R8) Although the problem is fundamentally different, our approach could be adapted to be used for estimating the causal effect of an exposure on outcome using many variants including invalid instrument with potentially correlated effects (i.e. violating the INSIDE assumption). Moreover, it can

be used in conjunction with other methods to estimate probabilities of valid instruments as we mentioned in the 'Discussion' (page 17):

"... The SH approach can be adapted in future to be used to identify the class of SNPs that show no pleiotropy (equivalent to class GI here), and the class that demonstrate pleiotropy (class G_P in this context). This approach would likely be robust to the the 'Instrument Strength Independent of Direct Effect' (InSIDE) assumption but may require the ZEMPA assumption."

Minor Comments

(C9) A question about presentation of simulation results. I remain a bit confused about when b differs across variants and when it does not. It seems that it would when β_{GP} and β_{GI} are correlated or in Sc.6. In these cases, is the average difference between estimated and true event bias averaged over snps as well as over simulations? Some clarification on this issue would help.

(R9) We have amended the text to explain the relationship between assumptions A1 (the 'InCLUDE' assumption) and A2 (the homogeneity in U) and our method (see the 'Underlying assumptions' in page 11):

"Our estimate of the adjustment factor, b_1 , is robust to violations of assumption A1. For a single SNP GI affecting incidence only, as in Figure 1(a), the genetic component of U is the entire shared genetic basis of I and P (i.e. common polygenic effect). Therefore, the genetic component of U is constant across all SNPs affecting incidence only (i.e. the class GI.). Since the non-genetic component of U is constant across all SNPs by definition, our estimate of the adjustment factor, b_1 , is robust to violations of assumption A2. However, the procedure of correcting bias for all variants assumes a constant confounder effect, U, across all variants (i.e. $b_1 = b_2$ in equation 8 which requires assumption A2 to hold ..."

We have clarified that estimated and true index event bias are only estimated once per simulation – i.e. not separately for each SNP by amending the last paragraph in page 12 to read:

"We performed 1000 simulations for each scenario and reported for each scenario the mean of the 1000 within-simulation differences between estimated adjustment factors and the true index event bias"

(C10) Page 8. Dudbridge et al argue that the Hedges-Olkin estimator has too large variance and use SIMEX instead. A comment about why the H-O estimator works here could be useful.

(R10) Hedges-Olkin is one of many possible methods to correct for the regression dilution. It is less preferable for analysing data with a small sample size as in the real data example presented in Dudbridge et al (The idiopathic pulmonary fibrosis (IPF) example), where the Hedges-Olkin gave an implausible result. In such situations, correcting the regression dilution using the SIMEX would more preferred than using the Hedges-Olkin, due to its increased variance. We have amended the text on page 9 to read:

“Due to its increased variance, using the Hedges-Olkin method to correct for the regression dilution would not be preferable when analysing data with a small sample size. In such situations, using an alternative method such as simulation extrapolation (SIMEX) could be more sensible”.

In our simulation studies, we compared our method with the method of Dudbridge et al with Hedges-Olkin and with simulation extrapolation for regression dilution. We have reported that in page 13:

“Results from the Slope-Hunter method (‘SH’ estimator) were compared with the unadjusted estimator and the estimator of the method of Dudbridge et al (2019) with Hedges-Olkin adjustment (DHO estimator) and with simulation extrapolation adjustment (DSIMEX estimator) for regression dilution. Because DHO and DSIMEX results were almost identical, we only reported the DHO results.”

(C11) It would be nice if the information in the tables could be displayed graphically.

(R11) As suggested, we have revised the presentation of our results and displayed graphically the estimated adjustment factors, Type-1 error rates and power. We have moved the tables with detailed results for all scenarios in the supplementary materials.

(C12) Discussion "whereby independent causes of the incidence become correlated when selecting only on cases and then may confound analysis of prognosis." Is hard to understand.

(R12) This statement has been rephrased (in page 16) to read:

“... whereby independent causes of incidence become correlated when selecting only on cases inducing bias in the analysis of outcome”.

(C13) It could be useful in motivating the problem and highlighting its significance to point out that this problem can occur essentially whenever a GWAS is conducted conditioning on a heritable trait and not all of these would typically be considered “prognosis”. For example, the problem could occur looking at waist to hip ratio conditional on BMI.

(R13) As suggested by the reviewer, we have defined the problem more generally and amended relevant text in the ‘Background’ section (page 2) to read:

“There is increasing interest in the use of genome wide association studies (GWAS) conditioned on a phenotype, such as a GWAS of blood pressure conditional on body mass index (BMI) so as to avoid only genetic variants associated with BMI appearing important. An example of such conditional analyses that is increasingly being used is GWAS of prognosis ... Such an analysis is referred to as ‘conditional analysis’ throughout this manuscript”

We have amended the paper title to reflect this:

“Slope-Hunter: A robust method for collider bias correction in conditional genome-wide association studies”

We have demonstrated the implementation of our approach by analysing data for fasting blood insulin levels conditional on BMI (see page 13).

Reviewer 2:

Mahmoud et al present a new method to correct association summary statistics of subsequent traits. The methodological foundation of the problem is solid and the proposed method seems logical. However, I found several weaknesses of this paper: (i) the paper difficult to read; (ii) the propose algorithm being quite ad hoc with many arbitrary choices (for strategies and parameters); (iii) the simulations are not presented clearly and the reader is left without a detailed constellation of parameters when and why would the SH method break down. Moreover, presenting no application to real data is a substantial weakness of the work and makes it much less appealing to a journal like Nature Communications, which has very broad readership. Below I proved a detailed list of questions and suggestions to improve the work and its presentation.

Major comments

14. *It is not clear how pruning is applied (if P-values are used to determine the order, which ones G-P or G-I) and whether it would introduce bias (winners curse).*

(R14) In our simulation study, we have not needed pruning because SNPs were simulated as independent variants. This is stated in ‘Simulations’ (page 12):

“We simulated four scenarios, each with 10,000 independent SNPs under Hardy-Weinberg equilibrium with minor allele frequencies drawn from a uniform distribution ...”.

For the real data analysis of “genetic factors causing fasting insulin independently of BMI”, added in the revised version as suggested by the reviewer, the pruning was randomly applied. This has been mentioned in page 13:

“... the pruning was randomly applied to avoid the winners curse bias problem”.

15. *While the general strategy is clear and logical, but the actual algorithm needs to be explained much clearer. Here are a few points that in my opinion need more clarification:*

(R15) We have amended the explanation of the algorithm. Our response to each of the points raised by the reviewer is as follows:

a. *“Exclusion of all SNPs assigned to the class G4 may discontinue the dimensional space” - I’m lost what it means.*

(R15.a) We have explained this by amending the relevant text in page 7 to read:

“Unlike G.. (G4 in the first version), the variants of the cluster G_i (G1 in the first version) are expected to form a linear pattern whose slope is the adjustment factor to correct the index event bias, see b_1 in equation 8. Our proposal aims to estimate this slope by identifying such a linear pattern. However excluding all SNPs assigned to the cluster G.. may distort such a pattern, e.g. by splitting it into two disconnected ellipses, leading to poor or invalid identification of the cluster ‘ G_i ’. ...”

b. "fitting a clustering solution f_j " – What is a clustering solution?

(R15.b) The text has been amended to read (page 8, step 3):

"We fit a cluster-based model f_j using ..."

c. What is the rationale behind the definition of ω_g ? Why is the second term not squared?

(R15.c) We have amended the text in page 7 to explain this: "This sub-sample is randomly selected using a weighted score ω_g , for each SNP $g \in G_{..}$ (lines 3-5). Since the index event bias is represented by the slope b_1 , equation 8, whose line is horizontal when there is no bias, variants with larger magnitudes of their vertical dimension's values are expected to be more informative if retained into the analysis among the $G_{..}$ class. Therefore, the weighting score is defined as a modified version of the Euclidean distance from the origin with larger influence for magnitudes of the conditional estimates, depicted against the Y-axis in the dimensional space. This implies that data points with larger distance from the origin, with higher weights given to their conditional estimates, hence the non-squared term under the root in the equation for ω_g (line 4), are more likely to be retained in the analysis."

d. The notations G_1, G_2, G_3, G_4 are not intuitive, please consider renaming these groups (e.g. G_I for incidence SNPs, G_{IP} for SNPs associated with both, etc.)

(R15.d) As suggested, we have renamed the notations: $G_1; G_2; G_3; G_4$ to be: G_I for SNPs affecting I only; G_{IP} for SNPs affecting both I and P ; G_P for SNPs affecting P only; $G_{..}$ for SNPs with neither effect on I nor P .

e. There are arbitrary parameters (δ , η), how robust is the method to variations in these values?

(R15.e) We have examined the performance of our method using different values of these input parameters and presented the results in Table S9 (supplementary materials). We have summarised these results in page 15:

"Table S9 shows ... Under each level of genetic correlation, different combination of the input values η and δ yielded similar results for all assessment measures ..."

f. In step 4 (page 8), it is not clear how the cluster centres are calculated (effect directions are arbitrary, aren't mean effects expected to be zero?)

(R15.f) We have amended the explanation of step 4 (now on page 8) to clarify that the two clusters are identified by the model f_j which estimates their means μ_I and μ_{IP} as points in the two-dimensional space. The Euclidean distance between these two mean points is then considered to calculate the value of the indicator H_j which sets to 1 if this Euclidean distance is less than a threshold. The relevant text in step 4 is now read:

"We define an indicator H_j which sets to 1 if the Euclidean distance between the means of the two clusters identified by the model f_j is not larger than ..."

g. What is the rationale for step 4, why situations where clusters G1, G2 that are far from each other are not considered?

(R15.g) Step 4 has been amended to explain this point:

“Variants of the cluster G_{IP} deviates from the linear pattern of cluster G_I based on their direct effects on P, see equation 8. Since these individual direct effects are small for polygenic traits, the means of the two clusters G_I and G_{IP} are expected to be close to each other. We define an indicator H_j which sets to 1 if the Euclidean distance between means of the two clusters identified by the model f_j is not larger than $\delta \cdot \min(s_{iv})$, where δ is a scalar set by the user (with a default value $\delta = 1$) and $\min(s_{iv})$ is the minimum marginal standard deviation across $i = G_I, G_{IP}$ clusters and $v = I, P$ traits. Otherwise H_j sets to zero (line 12). If the indicator H_j is 1, then clusters obtained from the model f_j are considered for estimating a candidate adjustment factor (slope) from the iteration j . The value of the scalar, δ , controls trimming of candidate estimates of the slope b_1 , with larger values of δ giving less conservative candidate estimates. For instance, if $\delta = 1$ (the default), then the cluster model f_j is considered as a candidate model only if the distance between means of its two mixture components is not larger than $\min(s_{iv})$.”

h. The pseudo-code is much clearer than the text description

(R15.h) We have amended the text description of the algorithm to make it clearer (page 7):

“Our approach, presented in Algorithm 1, starts by using the pruned GWAS statistics for the incidence I to obtain p-values for SNP-incidence associations, p_{GI} (line 1). A two-components bivariate mixture model (f) is then fitted ... to estimate membership probabilities, and assign variants, to the cluster $G_{..}$ (line 2). Since variants of this cluster affect neither I nor P, their pairs of association coefficients ... should have no structural function with respect to I or P. Therefore, their corresponding points plotted on the Cartesian coordinate system are expected to scatter around the origin in a noisy form with a high probability concentration since the majority of variants in GWAS studies are expected to belong to this cluster, Figure 3(b). We used a cluster-based model approach to fit the model f (line 2) in which each mixture component is defined as an ellipse with different geometric features: area denoted by ...; shape denoted by ...; orientation denoted by ..., that are determined by its estimated covariance matrix [16, 17], see Figure S2. The cluster $G_{..}$ is identified as the component including the variant with the smallest Euclidean distance from the origin.

Unlike $G_{..}$, the variants of the cluster G_I are expected to form a linear pattern whose slope is the adjustment factor to correct the index event bias, see b_1 in equation 8. Our proposal aims to estimate this slope by identifying such a linear pattern. However, excluding all SNPs assigned to the cluster $G_{..}$ may distort such a pattern, e.g. by splitting it into two disconnected ellipses, leading to poor or invalid identification of the cluster G_I . This issue is particularly problematic when the set of SNPs affecting only incidence has a relatively large number of variants with small effects on I as they are more likely to be misclassified to the $G_{..}$ cluster. To avoid distortion of the G_I pattern, which goes through the origin by definition (see equation 8), our method retains a fraction ... of the SNPs assigned to $G_{..}$. This sub-sample is randomly selected using a weighted score w_g , for each SNP ... (lines 3-5). Since the index event bias is represented by the slope b_1 , equation 8, whose line is horizontal when there is no bias, variants with larger magnitudes of their vertical dimension's values are expected to be more

informative if retained among the $G_{..}$ class. Therefore, the weighting score is defined as a modified version of the Euclidean distance from the origin with larger influence for magnitudes of the conditional estimates... This implies that data points with larger distance from the origin, with higher weights given to their conditional estimates, hence the non-squared term under the root in the equation for wg (line 4), are more likely to be retained in the analysis. All weights are then normalised to lie within $[0, 1]$ (line 6).

...”

16. The algorithm is not intuitive, why not fitting all (pruned) SNPs together at the same time using a 4 component bivariate Gaussian mixture model?

(R16) Our aim was to identify the pattern of the class G_1 (renamed to G_i in the revised version) rather than identifying memberships of the four classes. Equation (8) – that introduced the four potential variant statuses - does not represent our model of interest. It rather presents the mathematical forms of the conditional estimate defined by Dudbridge et. al. (2019) – shown in Equation (5) – corresponding to different statuses of variants in terms of their effects on I and P . The main purpose of the Equation (8) is to illustrate how consideration of only the class G_i - in estimating the adjustment factor - would be robust against the assumptions A1 (INCLUDE), that is required by the method of Dudbridge et. al. (2019).

Therefore, the accuracy of the G_4 (renamed $G_{..}$) and G_3 (renamed G_{p}) classifications were out of our scope because they do not suffer the index event bias. As a procedure for pre-processing data prior to our main analysis, we firstly fit a two-component mixture model ($G_{..}$ and $\hat{G}_{..}$) to exclude data points that scattered around the origin in a noisy form. We revised our text in page 7 to clarify that:

“... corresponding points plotted on the Cartesian coordinate system are expected to scatter around the origin in a noisy form with a high probability concentration since the majority of variants in GWAS studies are expected to belong to this cluster, Figure 3(b).”

Our simulated example, plotted in Figure (3), shows that our pre-processing procedure works well enough in identifying and removing the unaffected groups of variants $G_{..}$ and G_{p} from the data prior to our main analysis. However, a random subsample of $G_{..}$ data points is retained to facilitate pattern recognition of the target class (G_i).

17. I major weakness of the paper that is presents no application to real data, which would be extremely crucial to convince the readers that the formulated problem is important and the implications of the proposed correction are far-reaching.

(R17) As suggested, we have included an application to a published GWAS study of fasting blood insulin levels conditional on body mass index (BMI), by analysing data from this conditional analysis and from BMI GWAS. The data are described in detail in the ‘Methods’ (page 13) and the results are summarised in Table 3 and page 15 as follows:

“A GWAS meta-analysis of data from ... The risk alleles of the SNPs ‘rs7607980’ (the ‘COBLL1’ gene) and ‘rs1801282’ (the ‘PPARG’ gene) are associated with decreased BMI but increased BMI-adjusted blood insulin. These apparently paradoxical associations could arise from index event bias, given their significant associations with BMI, see Table 3.

We analysed the summary statistics using our adjustment procedure (SH) and Dudbridge’s method (DHO) with 12,792 LD-pruned SNPs. The estimated adjustment factor using the SH method was -0.136

(95% CI: -0.188 to -0.085). The adjustment factor estimated by the DHO estimator gave a result of -0.118. The negative slopes obtained by both adjustment methods implies that there are common causes of BMI and fasting insulin of concordant directions of effect. When we adjusted the conditional associations of each of the lead SNPs with the outcome using our method, the two paradoxical associations of COBLL1 (unadjusted conditional estimate and its standard error $\hat{\beta}(SE) = 0.030 (0.006)$) and PPARG (unadjusted $\hat{\beta}(SE) = 0.025 (0.004)$) were attenuated towards the null (adjusted $\hat{\beta}(SE) = 0.028 (0.006)$ and $0.023 (0.005)$ respectively). Similar results were obtained using the DHO method. Although these adjustments made little difference, there was an evidence of some confounding in the expected direction (i.e. inferred by estimated negative slopes) resulting in these coefficients were attenuated slightly towards the null, see Table 3.”

18. *An application would be particularly useful because one could see the impact of effect size bias on downstream analysis, such as MR.*

(R18) As suggested, we have added a real data application for genetic factors causing fasting insulin independently of BMI (page 13). In our response to the previous point (R17), we showed the unadjusted and adjusted associations for the lead variants (page 15):

“When we adjusted the conditional associations of each of the lead SNPs with the outcome using our method, the two paradoxical associations of COBLL1 (unadjusted conditional estimate and its standard error $\hat{\beta}(SE) = 0.030 (0.006)$) and PPARG (unadjusted $\hat{\beta}(SE) = 0.025 (0.004)$) were attenuated towards the null (adjusted $\hat{\beta}(SE) = 0.028 (0.006)$ and $0.023 (0.005)$ respectively). Similar results were obtained using the DHO method. Although these adjustments made little difference, there was an evidence of some confounding in the expected direction (i.e. inferred by estimated negative slopes) resulting in these coefficients were attenuated slightly towards the null, see Table 3”

19. *Tables 3-7 are very boring presentations of the results. Most of these should be move to the supplement and the authors should try to present some of these comparisons more succinctly as figures.*

(R19) This has been done as suggested by the reviewer.

20. *Figures 1-3 & 5 should be merged. I don't find any reference to Fig 8/9 in the text and do not find the plots very meaningful either. Either move it to supplement (and refer to it) or explain them much better and motivate their presence in the main text. I don't see much point of Figure 7 either, it does not provide too much insight into the hypothetical covariance structure of the effect estimates. Could it be coupled with Fig 4? Not sure why it should be a main figure. Figure 6 is great and very didactic.*

(R20) We have merged Figures 1-3 & 5 in a single figure (becomes Figure 1). We have replaced figures 8 & 9 with Figure S1 (placed in the supplement) and referred to it in the main text (Page 15):

"Figure S1 shows uncertainty about the assignment of variants to the cluster G_I , and highlights the misclassified variants in the cluster G_I under Sc.2".

We have moved figure 7 to the supplement (becomes Figure S2) as we think it provides a graphical presentation for the geometric parameters: area; shape; orientation. As suggested, we have kept Figure 6 (becomes Figure 3) in the main text.

21. *In the different scenarios (1-5), it is not clear which is the crucial parameter that leads to deteriorating performance of the SH method? G_1 explaining larger variance is clearly an advantage, but what role the polygenicity of incidence play here? Also, does the relative heritability of the G_1 and G_2 groups important? The authors provide little intuition in the Discussion.*

(R21) We have revised the simulated scenarios and added two more scenarios (named Sc.1 and Sc.3 in the revised version). Across Sc.1 – Sc.3, we hold equal number of variants for G_1 and G_2 (renamed as G_I and G_{IP} respectively) with different explained variation in incidence by each group, see Table 1.

In Sc.4, we increased both size and proportion of explained variation in I for the class G_{IP} violating the main underlying assumption of our approach (referred to as MAPCA, see the revised section 'underlying assumption' page 11).

Scenarios Sc.A1, Sc.A2 and Sc.A3 (that were called Sc.1, Sc.3 and Sc.5 in the first version respectively) were moved to the supplementary materials and described in Table S1 in which the class G_I has much fewer number of SNPs than in Sc.1 – Sc.3, with different explained variation in incidence by each group.

This enables to demonstrate the role of each simulation parameter at a time.

We have discussed roles of heritability of the clusters of variants in the 'underlying assumption' page 11:

"... By definition, variants that affect I and P via the same exposure, say E as depicted in Figure 1(c), should have the same value for Δ_g . This leads to an exact proportional relationship between their conditional estimates and their SNP-incidence association, typically like the key characteristic of the G_I class. The ability of our approach to estimate the true index event bias relies on the following fundamental assumption termed the Maximal Predictive Capability Assumption (MAPCA): the class G_I explains more variation in incidence than any subset of the class G_{IP} with the same Δ_g value".

22. *I don't understand how the SH method can maintain good control of the type I error even in scenario 5 and high genetic correlation, despite providing a poor estimate for the adjustment factor?*

(R22) Sc.5 has been renamed as 'Sc.A3' and moved to the supplementary materials. Under strong genetic correlation for this scenario, our method, SH, provides poor estimate for the adjustment factor as well as poor type-1 error rates (39.3% and 70.9% under correlation coefficients of -0.9 and 0.9 respectively) when averaged over SNPs with effect on incidence, for which there is an index event bias.

Type-1 error rates of all methods might be close to the nominal level, 0.05, when averaged over all SNPs as most SNPs have no bias. This has been explained for Sc.1-Sc.3 in 'Results' (page 14):

"the type-1 error rates obtained from the standard unadjusted analysis and from the adjusted analyses using (H-O) and our procedure (SH) are close to the nominal level, 0.05, when averaged over all SNPs; however the majority of SNPs have no index event bias. Among SNPs with effects on incidence, for which there is a bias, the type-1 error rates are inflated for the unadjusted analysis".

Minor comments

23. *The cluster parameters are mentioned strangely (parameters look like if they were a product) in the sentence starting "The geometric features of each cluster are determined by its covariance matrix that can be decomposed to the form of...". Also it has theta_k twice.*

(R23) This sentence has been rephrased (page 7):

"... each mixture component is defined as an ellipse with different geometric features: area denoted by ζ ; shape denoted by ϑ ; orientation denoted by θ that are determined by its estimated covariance matrix [16, 17], see Figure S2."

24. *Could the authors elaborate why their method is robust against sample overlap*

(R24) We have amended our text in the 'underlying assumption' (page 11) to clarify that: "Since any correlation between I and P should be included in U by definition [4], then our method, whose estimate of the adjustment factor is defined to capture the confounding effects, is robust against the overlap between samples in the conditional outcome GWAS and the incidence GWAS ..."

25. *"assuming that $b_1 \sim b_2$, which is typically the case for small effects as in GWAS" – Could the authors explain why?*

(R25) We have amended our text in the 'Underlying assumptions' (page 11) to explain that:

"For a single SNP G_I affecting incidence only, as in Figure 1(a), the genetic component of U is the entire shared genetic basis of I and P (i.e. common polygenic effect). Therefore, the genetic component of U is constant across all SNPs affecting incidence only (i.e. the class G_I). Since the non-genetic component of U is constant across all SNPs by definition, our estimate of the adjustment factor, b_1 , is robust to violations of assumption A2. However, the procedure of correcting bias for all variants assumes a constant confounder effect, U, across all variants (i.e. $b_1 = b_2$ in equation 8 which requires assumption A2 to hold. For a single SNP G_{IP} affecting both incidence and outcome, as in Figures 1(b) and 1(c), the SNP G_{IP} is itself a part of the common polygenic effect on I and P, i.e. contributing to the genetic component of U unless it is the SNP under consideration. Therefore when considering a single SNP G_{IP} , the genetic component of U is the residual common polygenic effect on I and P that equals the entire shared genetic basis of I and P, minus the component attributed to the SNP under consideration which is small compared with the total genetic effect for most polygenic traits. As a result, the genetic component of U is approximately constant across all SNPs, i.e. assumption A2 almost holds, for most polygenic traits."

26. Can the authors explain, why it was not necessary to vary the number of markers in the G3 group (and their explained variance)?

(R26) The variants of G3 group (renamed as G_p in the revised version) have no effects on incidence, so they do not suffer bias as illustrated in Figure 1(d). The simulation parameters were related to the number of markers with effect on incidence (and variation in 'incidence' explained by these markers). This applies only to the classes G_I and G_{IP}

Reviewers' Comments:

Reviewer #1:

Remarks to the Author:

The authors have made major revision efforts including adding an application to real data, extensive language revisions, and modifications to simulations. I find that most of my original comments are addressed satisfactorily. I have a few remaining questions/comments and some questions about the new material:

1. In equation 8, should the middle line for variants in G_{IP} be $\beta_{GP} + b_{2,g}\beta_{GI}$ to fully allow for a violation of A2, making the full requirement for the proposed method is that $b_{2,g} = b_1$ for all g ?

2. Two concerns about the estimate of variance of \hat{b}_1 :

2a. I am concerned that the step at line 17 of Alg 1 in which the estimate with the smallest estimated standard error is chosen leads to an underestimate of the se of \hat{b}_1 . For example, imagine that the true se of all \hat{b}_j is the same. Then clearly, this selection procedure will lead to an underestimate of the se. This may not be substantial, but the problem could be corrected by recomputing b_j for the selected value of λ . That is, select j^* to be the index corresponding to the estimator with minimum se. Then for j^* , repeat steps 9-14 sampling a new subset of variants. If this procedure doesn't change the estimate much then you have demonstrated that the variance of $s(\hat{b}_j)$ stemming from the sampling variation is too low for there to be much selection bias, and if not you will have a more unbiased estimate.

2b. I think that the calculation of $s(\hat{b}_j)$ doesn't include uncertainty in clustering or variance from sampling the λ proportion of G . . How does the value of $s(\hat{b}_j)$ compare to a variance estimate obtained by bootstrapping?

3. The choice of weights w_g seems arbitrary to me. Why square β_{GI} and not β'_{GP} ? It would be nice to see a justification for this choice.

4. Something that continues to bother me is that I would prefer to see a clear probabilistic model for the distribution of $\hat{\beta}_{GI}$ and $\hat{\beta}_{GP}$ and then have the estimation procedure stem from that model. Without a clearly stated model, it is hard to understand some of the algorithmic choices like the weights and the λ fraction of G ., (see point 3).

5. The statement in the discussion "Our analysis of BMI-adjusted fasting blood insulin suggests that an apparently paradoxical associations of the strong risk loci COBLL1 and PPARG with increased insulin levels may be partly due to collider bias, and that these genes may not have significant associations after adjustment" does not appear supported by the results. In the results, applying SH reduced the z-scores of these genes from 5 and 6.25 to 4.67 and 5.11 respectively. As observed in results, these are not dramatic changes and both loci have similar significance after adjustment. If the associations are indeed due to collider bias than SH has been unsuccessful at removing it.

6. Is it possible to test/assess MAPCA?

7. Since BMI and FG share so many variants, it seems possible that MAPCA is violated in this application. I think this possibility should at least be addressed and ideally tested or assessed empirically.

8. A general comment – the exposition is very long, a bit repetitive especially in the part describing the algorithm, and many sentences are still overly long and difficult to parse. Tightening up the writing could make the paper much easier to read.

Reviewer #2:

Remarks to the Author:

The authors have addressed several questions of mine and have made the manuscript clearer. This

also facilitated my comprehension of their method in more details, which however led to many more questions and doubts. My comments are of two kinds: further request to clean up of the method description and questions regarding the results and comparison with other methods/approaches.

A. I have several problems with the presentation and the details of the algorithm. Below I refer to the algorithm step in "[]" where my problem comes from.

[1] This is a very awkward notation and incorrect. It assumes that the effects are always positive. P-values are calculated rather as $2 * \Phi^{-1}(-\text{abs}(\beta/s))$.

Furthermore, later small "g" is used for individual SNP effects, now it is "G", i guess the formula refers to individual SNPs.

[2] The word "Gaussian" is missing from every mention of the mixtures in the whole manuscript.

[2] This equation has multiple problems:

1. This is only one of the EM steps (when the mixture proportions are fixed).

2. So far $G_{\{.,.\}}$ mean the class of null-null SNPs for I and P, now it is used as a Gaussian pdf function? $G'_{\{.,.\}}$ is never defined, the reader can only guess that it is the remaining component (which is a mix of I-only, P-only and PI SNPs), so this is a heterogeneous component, why would a single component bivariate Gaussian fit it well?

3. These kind of clustering is very difficult in our experience and does very not lead to the desired separation of SNPs - these could just separate negative from positive effects for example. The authors seem to ignore that effects can equally be positive and negative and most effects must be on average zero. Thus there is no point in fitting the means, they must be fixed at zero - no other value makes any sense (under any polygenic model) and they are just randomly fluctuating away from zero. Also, the $G_{\{.,.\}}$ component variances are known: for null SNPs the variances should be s_{GI}^2 and s_{GP}^2 and only the covariance is unknown and it depends on sample overlap and observational correlation of the traits.

[2] Parameters (zeta, theta) are defined only in the main text and unnecessarily complicated (the three parameters could simply be the two variances and one covariance term). Also, it seems that no cluster mean is fitted, but later (step 12) they compare cluster means, while an analogous model is fitted.

[9] This is not explained: is the selection probability proportional to ω_g ? This ω_g is an arbitrary measure, not based on any model: the probability of belonging to either I or IP group could also be defined as $1 - \exp(-\beta_I^2/se_I^2) - \exp(-\beta_P^2/se_P^2)$, why is the author's ω more justified? Based on what model?

[9] I do not follow the logic here: These null-null SNPs are only included to add a scatter at the origin to avoid two distinct clusters. But that could simply be avoided by forcing both cluster centres (for I and IP) to be (0,0). As I mentioned above, under most polygenic models, these SNP groups must have zero mean, since flipping alleles would otherwise the cluster centres.

[11] Since you added in the null-null SNPs, the slope will be a mix of the slope of the truly I-SNPs and the null SNPs (whose slope is driven by sample overlap). How does it bias the results?

[12] The introduction of H_j is completely unnecessary, can't you simply say that " $d^{\{j\}} \leq \delta * \min(s_{iv})$ "?

[12] Since both means should be zero, if selection is conducted properly, this extra filter would not be necessary. What is the rationale for this additional check?

[13] G_I SNPs are not a binary membership, but the model give a probability for each SNP to

belong to the I or IP component. What threshold is used to select G_I SNPs? Also, by design you mixed in G_{.,.} SNPs into the model, thus many of these will be also "predicted" to belong to the G_I component. What is the impact of including these SNPs in the regression on the slope estimation (in case the summary stats are coming from overlapping samples)?

B. The point of adjustment of one trait for another is because we believe that part of the genetic effects is mediated through I and it needs to be corrected for. When this correction is simply done by regression, the outcome trait is adjusted not for the causal effect of I on P, but rather the correlation between I and P, which leads to collider bias due to the confounder. So instead of doing a wrong adjustment, why not estimating the causal effect (α) I→P in a robust fashion (median/mode-based estimator) and then use this to model $P_{adj} = P - \alpha * I$ as an outcome trait (or if P is binary, include the $\alpha * I$ term in the logistic regression) and run a GWAS on it? This principle is followed in the software mtCOJO. Have the authors compared their correction to the one by mtCOJO [<https://www.nature.com/articles/s41467-017-02317-2> & software <https://cnsgenomics.com/software/gcta/#mtCOJO>]?

C. The MAPCA assumption does not seem to be too specific. Many slopes can exist on the β'_{GP} vs β_{GI} scatter plot, one which is due to the collider bias and others that are due to a genetically determined confounders (denoted by E by the authors). When they fit the two component Gaussian to the effect-pairs, the authors would decide that the one with the most SNPs in it corresponds to the collider bias part or they sum up the (squared) effect sizes to see which explains more variance. So the MAPCA assumption is not really specific, any assumption would do that makes any statement about the two sets of SNPs yielding two different slopes. This is reminiscent of the causal inference problem with correlated pleiotropy tackled by CAUSE (Morrison et al. 2020 - <https://pubmed.ncbi.nlm.nih.gov/32451458/>). So the question is why not first apply CAUSE to get the causal effect (a) from I to P and then $P_{adj} = P - \alpha * I$, as mentioned above, following "mtCOJO principle".

D. The application to real data confirms my worry that this proposed correction does not result in any noticeable correction for any adjusted effect, neither yield any different correction from the previously published version (Dudbridge et al 2019) for real data (Table 3). Without a clear demonstration of a meaningful difference of the new method over the older one and no indication that any of the adjusted effects suffer from collider bias, such an improved method presents very limited advance in the field. A very basic correction has been published (Ashard et al. 2015 - <https://pubmed.ncbi.nlm.nih.gov/25640676/>) long time ago and I do not see how this more advanced method could reveal anything that the old ones could not.

E. Do I understand correctly that simulation results presented in Table 2 (Sc 4) show that SH has higher type I error, more bias and lower power than the previous method from the authors (DHO)?

F. Finally, I wonder why the method is "sold" as an adjustment correction method instead of a causal effect estimation tool which is robust to correlated pleiotropy, which would be a far more interesting aspect. As long as an appropriate causal effect is estimated, proper trait correction can be applied as a corollary.

Point-by-point response

NCOMMS-20-04869A - Slope-Hunter: A robust method for collider bias correction in conditional genome-wide association studies

We are grateful to the reviewers, the senior editor (SE) and to the editor for the detailed and helpful comments and suggestions. In the following, our responses are in blue letters and the comments of the reviewers in black italic letters.

Reviewer 1:

The authors have made major revision efforts including adding an application to real data, extensive language revisions, and modifications to simulations. I find that most of my original comments are addressed satisfactorily. I have a few remaining questions/comments and some questions about the new material:

1. In equation 8, should the middle line for variants in G_{IP} be $\beta_{GP} + b_{2,g}\beta_{GI}$ to fully allow for a violation of A2, making the full requirement for the proposed method is that $b_{2,g} = b_1$ for all g ?

R1: As suggested by the reviewer, we have revised the relevant mathematical expression for variants G_{IP} in equation 8 (Equation 9 in the revised version) and amended accordingly the corresponding text in page 6:

“The estimated correction factor (\hat{b}_1) can then be used to correct bias for all SNPs, by substituting \hat{b} by \hat{b}_1 in Equation 7 assuming the confounder effects are constant across all SNPs (assumption A2) under which $b_1 = b_{2G}$ for all $G = G_{IP}$ in Equation 9.”

2. Two concerns about the estimate of variance of \hat{b}_1 :

2a. I am concerned that the step at line 17 of Alg 1 in which the estimate with the smallest estimated standard error is chosen leads to an underestimate of the se of \hat{b}_1 . For example, imagine that the true se of all \hat{b}_j is the same. Then clearly, this selection procedure will lead to an underestimate of the se. This may not be substantial, but the problem could be corrected by recomputing b_j for the selected value of λ . That is, select j^ to be the index corresponding to the estimator with minimum se. Then for j^* , repeat steps 9-14 sampling a new subset of variants. If this procedure doesn't change the estimate much then you have demonstrated that the variance of $s(\hat{b}_j)$ stemming from the sampling variation is too low for there to be much selection bias, and if not you will have a more unbiased estimate.*

R2a: As suggested by the reviewer 1 (comment 4), and the reviewer 2 (comment A), we have revised the algorithm's steps and defined a probabilistic model for the distribution of genetic associations with incidence (I) and prognosis (P). This model has then been used to derive the estimate of our correction factor (b_1). As suggested by the reviewer, the bootstrap approach has been implemented to estimate the standard error of \hat{b}_1 . The text in the last paragraph of page 7 now reads:

“The distributions of the observed associations for the variants affecting I, for which there is a collider bias are addressed as follows:

$$\begin{pmatrix} \hat{\beta}_{GI} \\ \hat{\beta}'_{GP} \end{pmatrix} \sim \pi_1^* N \left(\mathbf{0}, \begin{bmatrix} s_I^2 & b_1 s_I^2 \\ b_1 s_I^2 & b_1^2 s_I^2 \end{bmatrix} \right) + (1 - \pi_1^*) N \left(\mathbf{0}, \begin{bmatrix} s_I^2 & b_1 s_I^2 + \sigma_{IP} \\ b_1 s_I^2 + \sigma_{IP} & b_1^2 s_I^2 + \sigma_P^2 + b_1 \sigma_{IP} \end{bmatrix} \right), \quad (12)$$

where π_1^* represents the probability that a SNP G belongs to the cluster (G_I) affecting only I. We use the EM algorithm [15] to estimate the unknown parameters, b_1 , σ_{IP} and π_1^* (line 5). We use the Bootstrap estimation technique [16] to estimate standard error of the correction factor, $s(\hat{b}_1)$ (line 6)."

Following the reviewers' suggestions (Reviewer 1: comment 2, Reviewer 2: comments A(2) and A(3)), the revised algorithm has been substantially simplified and the number of required input parameters has been reduced. The parameter λ_j , which was defined (in the previous version) as the size of retained sub-sample from the identified cluster G_{..}, has been dropped. The only parameter remains required for our revised model setup is the p-value threshold for GWAS of I, that was represented by η (in the previous version) but it is renamed to λ in this revised version. The text in the first paragraph of page 7 now reads:

"Since the SNP-I associations are observed with no collider bias, in our context, then the SNPs G_P and G_{..} could be effectively identified by employing a p-value threshold in the study associated with I to exclude SNPs that are not associated with the trait (I)."

2b. I think that the calculation of $s(\hat{b}_j)$ doesn't include uncertainty in clustering or variance from sampling the λ proportion of G_{..}. How does the value of $s(\hat{b}_j)$ compare to a variance estimate obtained by bootstrapping?

R2b: As explained by our response R2a, we have implemented the bootstrapping approach to estimate the standard error of our correction factor, $s(\hat{b}_1)$. We have then used this to derive an estimate for the standard error of the adjusted genetic associations that include the uncertainty in estimating the correction factor. The second paragraph of page 8 has been revised to explain the relevant steps in the algorithm that are corresponding to this procedure:

"The standard error of the bias-adjusted associations is calculated as shown in line 9. The bias-adjusted estimates and their standard error are then returned for all SNPs (line 11)."

3. The choice of weights w_g seems arbitrary to me. Why square β_{GI} and not β'_{GP} ? It would be nice to see a justification for this choice.

R3: We have revised the algorithm and dropped the weights w_g that were assigned for the sub-sample retained from the identified cluster (G_{..}). Alternatively in this revised version, we excluded all SNPs with no effect on I, defined as those with a p-value greater than a threshold (λ) for their association with I. We described this by revising the text in the last paragraph of page 7 to read:

"The procedure starts by using the pruned GWAS statistics for I to calculate p-values of the SNP-I associations (line 2), and retain only the SNPs associated with I whose p-values less than a threshold (line 4). The distributions of the observed associations for the variants affecting I, for which there is a collider bias are addressed as follows: ..."

4. Something that continues to bother me is that I would prefer to see a clear probabilistic model for

the distribution of $\hat{\beta}_{GI}$ and $\hat{\beta}_{GP}$ and then have the estimation procedure stem from that model. Without a clearly stated model, it is hard to understand some of the algorithmic choices like the weights and the lambda fraction of $G_{..}$, (see point 3).

R4: As suggested by the reviewer, we have revised the ‘Methods’ section and defined a probabilistic model for the distribution of genetic associations with incidence (I) and prognosis (P). We have accordingly revised our estimation procedure (Methods: Motivating idea, pages 5-7) and presentation of our algorithm (Algorithm 1, page 8). The text of last paragraph in page 6 has been revised to clearly explain the model:

“Collider-bias correction using model-based clustering

Assuming the confounder effects are constant across all SNPs, the distributions of the SNP-I and SNP-P associations can be written under the proposed model in the form of

$$\begin{pmatrix} \beta_{GI} \\ \beta'_{GP} \end{pmatrix} \sim \pi_1 N \left(\underline{0}, \begin{bmatrix} \sigma_I^2 & b_1 \sigma_I^2 \\ b_1 \sigma_I^2 & b_1^2 \sigma_I^2 \end{bmatrix} \right) + \pi_2 N \left(\underline{0}, \begin{bmatrix} \sigma_I^2 & & b_1 \sigma_I^2 + \sigma_{IP} \\ b_1 \sigma_I^2 + \sigma_{IP} & b_1^2 \sigma_I^2 + \sigma_P^2 + b_1 \sigma_{IP} \end{bmatrix} \right) \\ + \pi_3 \begin{pmatrix} \eta_0 \\ N(0, \sigma_P^2) \end{pmatrix} + \pi_4 \begin{pmatrix} \eta_0 \\ \eta_0 \end{pmatrix}, \quad (10)$$

where π_1, π_2, π_3 and π_4 denote the probabilities that a SNP belongs to the clusters $G_{I.}$, G_{IP} , $G_{.P}$ and $G_{..}$, defined in Equations 8a-8d, respectively, with $\sum_{k=1..4} \pi_k = 1$, whereas $\underline{0}$ is a 2×1 zero-vector and η_0 is the probability point mass at 0. The latter two components in the model, shown in Equation 10, represent clusters ($G_{.P}$ and $G_{..}$) that do not affect I, hence do not suffer from the collider bias, and are then uninformative for our analysis. Since the SNP-I associations are observed with no collider bias, in our context, then the SNPs $G_{.P}$ and $G_{..}$ could be effectively identified by employing a p-value threshold in the study associated with I to exclude SNPs that are not associated with the trait (I).”

5. The statement in the discussion “Our analysis of BMI-adjusted fasting blood insulin suggests that an apparently paradoxical associations of the strong risk loci *COBLL1* and *PPARG* with increased insulin levels may be partly due to collider bias, and that these genes may not have significant associations after adjustment” does not appear supported by the results. In the results, applying SH reduced the z-scores of these genes from 5 and 6.25 to 4.67 and 5.11 respectively. As observed in results, these are not dramatic changes and both loci have similar significance after adjustment. If the associations are indeed due to collider bias than SH has been unsuccessful at removing it.

R5: We have applied our revised method to the same data example and reported the obtained results in Table 2. The revised results show that applying the SH method reduced the z-scores of the genes *COBLL1* and *PPARG* from 5.00 and 6.25 to 4.17 and 3.80 respectively. Since the changes in the z-scores depend on both the estimated correction factor and the effect of the genetic variants on BMI, we have additionally applied the adjustment methods on the associations of the *FTO* gene (as it has distinctly larger effect on BMI). In Table 2, we show that applying SH and DHO methods changed the z-score of the *FTO* gene from 0.25 to -6.00 and to -1.5 respectively. We have revised the text in last paragraph of page 13 to read:

“The risk alleles of the SNPs rs7607980 (the *COBLL1* gene) and rs1801282 (the *PPARG* gene) were associated with decreased BMI but increased FI. These apparently paradoxical associations could arise

from collider bias, given their associations with BMI (Table 2). The variant rs1421085 (gene FTO) had the strongest association with BMI, but appeared to be not associated with FI. The lack of association with FI could also arise from the collider bias, given the strong association with BMI.”,

the first paragraph of page 14 to read:

“We analysed the GWAS summary statistics for BMI and fasting blood insulin conditional on BMI (FI) using the DHO method and the Slope-Hunter method. The adjustment factor estimated using the Slope-Hunter method was -0.317 (95% CI: -0.417 to -0.218 based on a standard error of 0.051 estimated using 10,000 bootstrap samples), whereas the DHO method gave an estimate of -0.118, which slightly changed to -0.113 (95% CI: -0.151 to -0.084) when corrected for regression dilution using DSIMEX [5]. The adjustment factors obtained by all adjustment methods are negative implying that there are common causes of BMI and fasting blood insulin of concordant net directions of effect. Under the adjusted analyses, the two apparently paradoxical associations of COBLL1 and PPARG were attenuated towards the null, with greater attenuations under the Slope-Hunter method (Table 2). Our adjustment revealed an association for the FTO gene with fasting blood insulin in a direction that is concordant with the direction of its association with BMI. The same direction of association was obtained using the DHO method, but the estimated coefficient was closer to the null.”,

and the last paragraph of page 15 (Discussion) to read:

“Our analysis of BMI-adjusted fasting blood insulin (FI) suggests that apparently paradoxical associations of the strong risk loci COBLL1 and PPARG with increased insulin levels may be partly due to collider bias, and that these associations have been attenuated towards the null after adjustment. It has been suggested that risk alleles of the COBLL1 and PPARG genes have considerable associations with BMI [19], and this could lead to biased association when conditioning on BMI, due to collider bias [13]. The association of another strong BMI risk loci (FTO gene) with the outcome (FI) after correction showed a strong association in the same direction as its association with BMI. Our findings suggest that the common causes of BMI and insulin levels, the source of the collider bias, have effects on both traits with concordant directions. We have presented the results of six variants that were either strongly associated with BMI or associated with the outcome before correction. The concordant directions, identified by our analysis, are in line with the observed association between insulin resistance and obesity [23, 24] and agrees with the adjustment factor estimated using alternative methods [5].”

6. Is it possible to test/assess MAPCA?

R6: We have revised the definition of this assumption and expressed it mathematically in terms of the residuals ($e_{\{G\}} = \beta_{\{GP\}}^{\{\prime\}} - b_{\{1\}} \cdot \beta_{\{GI\}}$), see “Underlying assumptions” (page 9). Therefore, we renamed the assumption to be ‘Zero Modal Residual Assumption ‘ZEMRA’ (ZEMRA). This assumption is defined such that the largest number of similar individual-SNP ratios for SNP-prognosis to SNP-incidence associations comes from the class of SNPs only affecting incidence, even if the majority of SNPs have direct effects on both incidence and prognosis. Since the true clusters of each SNP are unknown, the ZEMRA is not a testable assumption. We have amended the text in the first paragraph of the Discussion (page 15) to read:

“Our analytic approach assumes the analysed SNPs are independent, do not interact with the confounders, have linear effects on I and outcome, and have no interaction with I. Moreover, it requires Zero Modal Residual Assumption (ZEMRA) that resembles the ZEMPA assumption for the MR analysis, but with respect to the residuals ($e_{\{G\}} = \beta_{\{GP\}}^{\{\prime\}} - b_{\{1\}} \cdot \beta_{\{GI\}}$) rather than

pleiotropy. The ZEMRA, like ZEMPA, is generally not a testable assumption since the true clusters of all SNPs are usually unknown.”

7. Since BMI and FG share so many variants, it seems possible that MAPCA is violated in this application. I think this possibility should at least be addressed and ideally tested or assessed empirically.

R7: We agree with the reviewer that BMI and fasting blood insulin share many variants, that are likely to have correlated effects on both traits. If this is the case, then this should violate the InCLUDE assumption required by the DHO method. As demonstrated by our simulation studies, the Slope-Hunter method is robust against violation of the 'INCLUDE' assumption. ZEMRA, the underlying assumption of the Slope-Hunter method, assumes that the largest number of similar ratios for SNP-prognosis to SNP-incidence associations comes from the true cluster G_I , even if the majority of SNPs have direct effects on both incidence and prognosis (cluster G_{IP}). In order for the ZEMRA assumption to be violated, there would need to be a single exposure that caused both BMI and fasting blood insulin - in this context - and was associated with more SNPs than the number associated with BMI only. As demonstrated in our response (R6) this assumption is generally not testable.

In this revised version, we have additionally applied our method to GWAS of a case-only study of breast cancer mortality, in which there are likely shared genetic pathways between the incidence (breast cancer risk) and prognosis (breast cancer mortality). The results obtained by both adjustment methods (Slope-Hunter and DHO) are compared in Table 3, and we have added a relevant paragraph in page 14 whose text reads:

“Differences between DHO and slope-hunter could be explained by potential violations to the InCLUDE assumption, a key assumption for the DHO method, as breast cancer incidence and mortality are likely to share genetic pathways that may result in correlated effects on both incidence and prognosis.”

8. A general comment – the exposition is very long, a bit repetitive especially in the part describing the algorithm, and many sentences are still overly long and difficult to parse. Tightening up the writing could make the paper much easier to read.

R8: As suggested by the reviewer, we have amended the manuscript by rephrasing the overly long sentences throughout and tightened the writing up.

Reviewer 2:

The authors have addressed several questions of mine and have made the manuscript clearer. This also facilitated my comprehension of their method in more details, which however led to many more questions and doubts. My comments are of two kinds: further request to clean up of the method description and questions regarding the results and comparison with other methods/approaches.

A. I have several problems with the presentation and the details of the algorithm. Below I refer to the algorithm step in “[]” where my problem comes from.

RA: As suggested by the reviewer, we have revised the algorithm and amended its presentation. We list our revisions below, against each corresponding points made by the reviewer.

[1] This is a very awkward notation and incorrect. It assumes that the effects are always positive. P-values are calculated rather as $2 * \Phi^{-1}(-\text{abs}(\beta/s))$. Furthermore, later small "g" is used for individual SNP effects, now it is "G", i guess the formula refers to individual SNPs.

R[1]: We agree with the reviewer. We have revised this and amended the mathematical expression and the notations, as suggested, so that the first two lines of the algorithm now read:

$$\begin{aligned} &1: \text{ for all } g \in G \text{ do} \\ &2: \quad p_{gI} = p \left(\chi_{(1)}^2 > \left(\frac{\hat{\beta}_{gI}}{s_{gI}} \right)^2 \right) \end{aligned}$$

[2] The word "Gaussian" is missing from every mention of the mixtures in the whole manuscript.

R[2]a: We have now revised this and clearly described the mixture components using Gaussian distributions. For example, the text of the third paragraph in page 7 reads:

“In the following, we propose a correction factor estimation procedure that is computationally simple and relies on model-based clustering to identify the cluster (G_I.) with a proportional relationship between the SNP-I and SNP-P associations. Our procedure solves a clustering problem in 2-dimensional space using a bivariate Gaussian mixture model of the effect-size distributions ...”.

Moreover, all the mathematical representations for the mixture models imply using the Gaussian distributions, as shown in Equation (12) for example:

$$\begin{pmatrix} \hat{\beta}_{GI} \\ \hat{\beta}'_{GP} \end{pmatrix} \sim \pi_1^* N \left(\underline{0}, \begin{bmatrix} s_I^2 & b_1 s_I^2 \\ b_1 s_I^2 & b_1^2 s_I^2 \end{bmatrix} \right) + (1 - \pi_1^*) N \left(\underline{0}, \begin{bmatrix} s_I^2 & b_1 s_I^2 + \sigma_{IP} \\ b_1 s_I^2 + \sigma_{IP} & b_1^2 s_I^2 + \sigma_P^2 + b_1 \sigma_{IP} \end{bmatrix} \right), \quad (12)$$

[2] This equation has multiple problems:

1. This is only one of the EM steps (when the mixture proportions are fixed).

R1: This equation has been removed from the algorithm and replaced by the Equation (12) shown as part of our response to the previous point (R[2]a).

2. So far $G_{\{.,.\}}$ mean the class of null-null SNPs for I and P, now it is used as a Gaussian pdf function? $G'_{\{.,.\}}$ is never defined, the reader can only guess that it is the remaining component (which is a mix of I-only, P-only and PI SNPs), so this is a heterogeneous component, why would a single component bivariate Gaussian fit it well?

R2: We agree with the reviewer that the notations used in the equation (presented in the previous version at line 2 of the algorithm) were not ideal. Therefore, we have revised this and defined the Equation (12) in this revised version.

We have revised our method to exclude the SNPs which do not affect I (null SNPs) using a p-value threshold (denoted by λ in this revised version) for their associations with I. Therefore, there is no null component (as was previously remained from the set $G_{..}$) retained in the analysis anymore. This has simplified the estimation procedure and facilitated fitting a two-component mixture model, as presented in Equation (12), for two homogeneous components ($G_{.I}$) and (G_{IP}).

3. These kind of clustering is very difficult in our experience and does very not lead to the desired separation of SNPs - these could just separate negative from positive effects for example. The authors seem to ignore that effects can equally be positive and negative and most effects must be on average zero. Thus there is no point in fitting the means, they must be fixed at zero - no other value makes any sense (under any polygenic model) and they are just randomly fluctuating away from zero. Also, the $G_{\{.,.\}}$ component variances are known: for null SNPs the variances should be s_{GI}^2 and s_{GP}^2 and only the covariance is unknown and it depends on sample overlap an observational correlation of the traits.

R3: As suggested by the reviewer, we have revised our definitions for the model-based clustering. As presented in the 'Motivating idea' (page 5):

"We assume a SNP (G) can belong to one of four mutually exclusive clusters according to its effects on the traits I and P: ..."

The following paragraphs have been revised to read (first paragraph in page 6):

"The SNPs in the first two clusters ($G_{.I}$ and G_{IP}) have non-zero bias terms, $\beta'_{GP} - \beta_{GP} \neq 0$, whose magnitude is proportional to their effects on I, see Equation 4. SNPs of the second cluster (G_{IP}) have potential correlated effects on I and P. This allows violation of the INCLUDE assumption (A1) formulated by Dudbridge et. al. (2019), as we allow $\sigma_{IP} \neq 0$ (Equation 8b). The SNPs in the third ($G_{.P}$) and fourth ($G_{..}$) clusters are not associated with I, hence they do not suffer bias, i.e. $\beta'_{GP} = \beta_{GP}$. Consequently, we reformulate Equation 4 ..."

The next paragraph in page 6 has also been revised to read:

"Instead of regressing $\hat{\beta}'_{GP}$ on $\hat{\beta}_{GI}$ for all SNPs, as implemented in alternative methods [5], we propose modelling the bivariate distribution of the effect-sizes using a Gaussian model-based clustering technique from which the cluster of $G_{.I}$ SNPs can be identified, and then used for estimating the correction factor (b_1). ..."

We have expressed our mixture model as a mixture of Gaussian components, with fixed zero means:

$$\begin{pmatrix} \hat{\beta}_{GI} \\ \hat{\beta}'_{GP} \end{pmatrix} \sim \pi_1^* N \left(\underline{0}, \begin{bmatrix} s_I^2 & b_1 s_I^2 \\ b_1 s_I^2 & b_1^2 s_I^2 \end{bmatrix} \right) + (1 - \pi_1^*) N \left(\underline{0}, \begin{bmatrix} s_I^2 & b_1 s_I^2 + \sigma_{IP} \\ b_1 s_I^2 + \sigma_{IP} & b_1^2 s_I^2 + \sigma_P^2 + b_1 \sigma_{IP} \end{bmatrix} \right), \quad (12)$$

We have then estimated the unknown parameters: b_1 (the correction factor, which is the slope of the first cluster); σ_{IP} (the covariance between I and P); π_1^* (probability that a SNP G belongs to the cluster G_I).

[2] Parameters (ζ , θ) are defined only in the main text and unnecessarily complicated (the three parameters could simply be the two variances and one covariance term). Also, it seems that no cluster mean is fitted, but later (step 12) they compare cluster means, while an analogous model is fitted.

R[2]: This previous representation of the model-based cluster has been revised following reviewers' suggestions (Reviewer 1: comment 2, Reviewer 2: comments A(2) and A(3)). The revised algorithm has been substantially simplified and the input parameters of interest are redefined as explained in the previous point (R3), and as shown in Equation (12): b_1 (the correction factor, which is the slope of the first cluster); σ_{IP} (the covariance between I and P); π_1^* (probability that a SNP G belongs to the cluster G_I).

[9] This is not explained: is the selection probability proportional to ω_g ? This ω_g is an arbitrary measure, not based on any model: the probability of belonging to either I or IP group could also be defined as $1 - \exp(-\beta_I^2 / se_I^2) - \exp(-\beta_P^2 / se_P^2)$, why is the author's ω more justified? Based on what model?

R[9]a: We have revised the algorithm and dropped the weights ω_g that were assigned for the sub-sample retained from the identified cluster ($G_{..}$). In this revised version, we excluded all SNPs with no effect on I, defined as those with a p-value greater than a threshold (λ) for their association with I. We described this by revising the text in the last paragraph of page 7 to read:

“The procedure starts by using the pruned GWAS statistics for I to calculate p-values of the SNP-I associations (line 2), and retain only the SNPs associated with I whose p-values are less than a threshold (line 4). The distributions of the observed associations for the variants affecting I, for which there is a collider bias are addressed as follows: ...”

[9] I do not follow the logic here: These null-null SNPs are only included to add a scatter at the origin to avoid two distinct clusters. But that could simply be avoided by forcing both cluster centres (for I and IP) to be (0,0). As I mentioned above, under most polygenic models, these SNP groups must have zero mean, since flipping alleles would otherwise the cluster centres.

R[9]b: As suggested by the reviewer, we have defined the distribution of the genetic associations as a mixture of bivariate Gaussian with means of (0,0). Equation (12) and the last paragraph in page 7 explain that:

“The distributions of the observed associations for the variants affecting I, for which there is a collider bias are addressed as follows:

$$\begin{pmatrix} \hat{\beta}_{GI} \\ \hat{\beta}'_{GP} \end{pmatrix} \sim \pi_1^* N \left(\underline{0}, \begin{bmatrix} s_I^2 & b_1 s_I^2 \\ b_1 s_I^2 & b_1^2 s_I^2 \end{bmatrix} \right) + (1 - \pi_1^*) N \left(\underline{0}, \begin{bmatrix} s_I^2 & b_1 s_I^2 + \sigma_{IP} \\ b_1 s_I^2 + \sigma_{IP} & b_1^2 s_I^2 + \sigma_P^2 + b_1 \sigma_{IP} \end{bmatrix} \right), \quad (12)$$

...”.

[11] Since you added in the null-null SNPs, the slope will be a mix of the slope of the truly I-SNPs and the null SNPs (whose slope is driven by sample overlap). How does it bias the results?

R[11]: As explained in previous points (R[9]) we have revised the method and excluded all SNPs with no effect on I, defined as those with a p-value greater than a threshold (λ) for their association with I. We described this by revising the text in the last paragraph of page 7 to read:

“The procedure starts by using the pruned GWAS statistics for I to calculate p-values of the SNP-I associations (line 2), and retain only the SNPs associated with I whose p-values less than a threshold (line 4). The distributions of the observed associations for the variants affecting I, for which there is a collider bias are addressed as follows: ...”

[12] The introduction of H_j is completely unnecessary, can't you simply say that " $d^{(j)} \leq \delta \cdot \min(s_{iv})$ "?

R[12]a: As suggested by the reviewer, we have completely removed this step as it became not necessary anymore because we set the means of both clusters (G_I and G_{IP}) to be zero.

[12] Since both means should be zero, if selection is conducted properly, this extra filter would not be necessary. What is the rationale for this additional check?

R[12]b: As responded to the previous point, we have completely removed this step as it became not necessary anymore because we set the means of both clusters (G_I and G_{IP}) to be zero.

[13] G_I SNPs are not a binary membership, but the model give a probability for each SNP to belong to the I or IP component. What threshold is used to select G_I SNPs? Also, by design you mixed in $G_{\{.,.\}}$ SNPs into the model, thus many of these will be also "predicted" to belong to the G_I component. What is the impact of including these SNPs in the regression on the slope estimation (in case the summary stats are coming from overlapping samples)?

R[13]: The estimation procedure of our revised method derives the correction factor estimate \hat{b}_1 based on the variance-covariance matrix of the first mixture component in Equation 12. Since the fitted model includes only two clusters, the SNP is implicitly assigned to the cluster for which its probability is larger than 0.5, e.g. the red and grey points in Figures 3(b) and 3(d) are the SNPs assigned to the cluster G_I and G_{IP} respectively.

B. The point of adjustment of one trait for another is because we believe that part of the genetic effects is mediated through I and it needs to be corrected for. When this correction is simply done by regression, the outcome trait is adjusted not for the causal effect of I on P, but rather the correlation between I and P, which leads to collider bias due to the confounder. So instead of doing a wrong adjustment, why not estimating the causal effect (α) $I \rightarrow P$ in a robust fashion (median/mode-based estimator) and then use this to model $P_{adj} = P - \alpha * I$ as an outcome trait (or if P is binary, include the $\alpha * I$ term

in the logistic regression) and run a GWAS on it? This principle is followed in the software mtCOJO. Have the authors compared their correction to the one by mtCOJO [<https://www.nature.com/articles/s41467-017-02317-2> & software <https://cnsgenomics.com/software/qcta/#mtCOJO>]?

R(B): This suggestion is applicable when the data can identify a causal effect from I to P. However, as noted in the Introduction, we are also interested case-only studies of disease progression, in which P is only defined where binary I=1. Then we cannot identify a causal effect from I to P, and the Mendelian randomisation (MR) framework cannot be used. Instead, we are estimating alpha as the path from I to P through U, induced by conditioning on I. We show that in linear models, $P_{adj} = P - \alpha * I$ where P is now conditional on I and alpha is the regression of these P effects on I effects (Dudbridge et al. 2019). mtCOJO could actually be used in this way, although it is not obvious from its published description.

With quantitative I, our approach is also useful if the summary statistics for P are only available conditional on I (as in the example of GWAS of BMI-adjusted fasting blood insulin, Table 2), whereas the mtCOJO approach would be appropriate if the statistics for P are not conditioned on I but we want the direct effects on P. There is further discussion on the relationship between our underlying set-up and MR in a forthcoming paper by Bowden

<https://www.medrxiv.org/content/10.1101/2020.10.20.20216358v1.full>

It is true that we are using SNPs as instrumental variables, and our approach is similar to the mode-based estimators in MR, which we now make explicit with our ZEMRA assumption (see 'Underlying assumption', page 9). The mtCOJO approach is analogous to inverse-variance weighted MR in terms of its assumptions (and the earlier approach by Dudbridge et al. (2019) to MR-Egger).

We appreciate the reviewer raising this interesting point and have added the following to the Introduction (page 3):

“When conditioning on a quantitative trait, the direct SNP-outcome associations could be obtained by using Mendelian randomisation to estimate the causal effect of I on P and then subtracting the $G \rightarrow I \rightarrow P$ path from the total $G \rightarrow P$ association. This approach, implemented in the mtCOJO software, presumes a causal effect of I on P and the availability of unconditional $G \rightarrow P$ effects, but is not applicable to case-only studies of disease progression.”

*C. The MAPCA assumption does not seem to be too specific. Many slopes can exist on the β'_{GP} vs β_{GI} scatter plot, one which is due to the collider bias and others that are due to a genetically determined confounders (denoted by E by the authors). When they fit the two component Gaussian to the effect-pairs, the authors would decide that the one with the most SNPs in it corresponds to the collider bias part or they sum up the (squared) effect sizes to see which explains more variance. So the MAPCA assumption is not really specific, any assumption would do that makes any statement about the two sets of SNPs yielding two different slopes. This is reminiscent of the causal inference problem with correlated pleiotropy tackled by CAUSE (Morrison et al. 2020 - <https://pubmed.ncbi.nlm.nih.gov/32451458/>). So the question is why not first apply CAUSE to get the causal effect (a) from I to P and then $P_{adj} = P - \alpha * I$, as mentioned above, following “mtCOJO principle”.*

R(C): The definition of our assumption has been amended and clarified using mathematical expressions. We have renamed it from MAPCA to ZERo Modal Residual Assumption (ZEMRA) to reflect its expression in terms of the residuals ($e_{G} = \beta_{GP}' - b_1; \beta_{GI}$), see “Underlying

assumptions” (page 9). This assumes the largest number of similar individual-SNP ratios for SNP-prognosis to SNP-incidence associations comes from the class of SNPs only affecting incidence, even if the majority of SNPs have direct effects on both incidence and prognosis. The G_I cluster and its corresponding slope (b_1) is identified as described in page 7:

“In the following, we propose a ... The true correction factor (b_1) maximises the density of points lying on, or scattered ‘closely’ around, the line $\hat{\beta}^{\prime}_{GP} = b_1 \hat{\beta}_{GI}$, i.e. points for which there is a proportional relationship between SNP-I and SNP-P associations (the key characteristic underlying the true cluster of SNPs, G_I , affecting only I), ...”.

Genetic variants of other clusters do not generally have a proportional relationship between their SNP-P and their SNP-I associations. Therefore, they would not have the same slope, unless the whole set of their effects on I and P act through the same exposure (E). If there is a set of SNPs affecting both I and P via the same exposure E, with larger set than the cluster G_I , then the ZEMRA is violated leading to misidentification of the true cluster.

As explained in the previous point (R(B)), the suggestion of applying the CAUSE method is applicable when the data can identify a causal effect from I to P. However, as noted in the Introduction, we are also interested case-only studies of disease progression, in which P is only defined where binary $I=1$. Then we cannot identify a causal effect from I to P, and the Mendelian randomisation (MR) framework cannot be used.

D. The application to real data confirms my worry that this proposed correction does not result in any noticeable correction for any adjusted effect, neither yield any different correction from the previously published version (Dudbridge et al 2019) for real data (Table 3). Without a clear demonstration of a meaningful difference of the new method over the older one and no indication that any of the adjusted effects suffer from collider bias, such an improved method presents very limited advance in the field. A very basic correction has been published (Ashard et al. 2015 - <https://pubmed.ncbi.nlm.nih.gov/25640676/>) long time ago and I do not see how this more advanced method could reveal anything that the old ones could not.

R(D): Following the reviewer comments, the revised method has been applied on the same real data example (BMI-adjusted fasting insulin) and on a new case-only study of breast cancer mortality. Results of these two examples are presented in Tables 2 and 3 respectively.

The revised results for the BMI-adjusted fasting insulin show that applying the SH method reduced the z-scores of the genes COBLL1 and PPARG from 5.00 and 6.25 to 4.17 and 3.80 respectively, whereas the DHO adjustments reduced them to 4.67 and 4.60 respectively. Since the changes in the z-scores depend on both the correction factor, estimated by the adjustment method, and the effect of the genetic variants on BMI, we have (additionally in this revised version) applied the adjustment methods to the associations of the FTO gene (as it has distinctly larger effect on BMI). In Table 2, we show that applying SH and DHO methods changed the z-score of the FTO gene from 0.25 to -6.00 and to -1.5 respectively. We have revised the text in last paragraph of page 13 to read:

“The risk alleles of the SNPs rs7607980 (the COBLL1 gene) and rs1801282 (the PPARG gene) were associated with decreased BMI but increased FI. These apparently paradoxical associations could arise from collider bias, given their associations with BMI (Table 2). The variant rs1421085 (gene FTO) had the strongest association with BMI, but appeared to be not associated with FI. The lack of association with FI could also arise from collider bias, given the strong association with BMI.”,

the first paragraph of page 14 to read:

“We analysed the GWAS summary statistics for BMI and fasting blood insulin conditional on BMI (FI) using the DHO method and the Slope-Hunter method. The adjustment factor estimated using the Slope-Hunter method was -0.317 (95% CI: -0.417 to -0.218 based on a standard error of 0.051 estimated using 10,000 bootstrap samples), whereas the DHO method gave an estimate of -0.118, which slightly changed to -0.113 (95% CI: -0.151 to -0.084) when corrected for regression dilution using DSIMEX [5]. The adjustment factors obtained by all adjustment methods are negative implying that there are common causes of BMI and fasting blood insulin of concordant net directions of effect. Under the adjusted analyses, the two apparently paradoxical associations of COBLL1 and PPARG were attenuated towards the null, with greater attenuations under the Slope-Hunter method (Table 2). Our adjustment revealed an association for the FTO gene with fasting blood insulin in a direction that is concordant with the direction of its association with BMI. The same direction of association was obtained using the DHO method, but estimated coefficient was closer to the null.”,

and the last paragraph of page 15 (Discussion) to read:

“Our analysis of BMI-adjusted fasting blood insulin (FI) suggests that apparently paradoxical associations of the strong risk loci COBLL1 and PPARG with increased insulin levels may be partly due to collider bias, and that these associations have been attenuated towards the null after adjustment. It has been suggested that risk alleles of the COBLL1 and PPARG genes have considerable associations with BMI [19], and this could lead to biased association when conditioning on BMI, due to collider bias [13]. The association of another strong BMI risk loci (FTO gene) with the outcome (FI) could be corrected suggesting a strong association in the same direction as its association with BMI. Our findings suggest that the common causes of BMI and insulin levels, the source of the collider bias, have effects on both traits with concordant directions. We have presented the results of six variants that were either strongly associated with BMI or associated with the outcome before correction. The concordant directions, identified by our analysis, are in-line with the observed association between insulin resistance and obesity [23, 24] and agrees with the adjustment factor estimated using alternative methods [5].”

The differences between adjustments made by both methods (SH and DHO) were more pronounced when correcting GWAS of the case-only study of breast cancer mortality (Table 3), in which there are likely shared genetic pathways between the incidence (breast cancer risk) and prognosis (breast cancer mortality). we have added a relevant paragraph addressing this in page 14 whose text reads:

“... Differences between DHO and slope-hunter could be explained by potential violations to the InCLUDE assumption, a key assumption for the DHO method, where breast cancer incidence and mortality are likely to share genetic pathways that may result in correlated effects on both incidence and prognosis.”

E. Do I understand correctly that simulation results presented in Table 2 (Sc 4) show that SH has higher type I error, more bias and lower power than the previous method from the authors (DHO)?

R[E]: This was not correct. The Table 2 (Sc4) in the previous version was presented as follows:

Table 2: Means and standard deviations (SD) of the differences between estimated adjustment factors using H-O and Slope-Hunter methods, and the true index event bias (B) over 1000 simulations of 10,000 independent SNPs, conditional on incidence as a quantitative trait, for five scenarios (Sc.1 - Sc.5) described in the first and second rows

		% G_1 SNPs (proportion of variation in incidence explained by G_1) Vs. % G_2 SNPs (proportion of variation in incidence explained by G_2)									
		Sc.1: 1% (0.45) Vs. 9% (0.05)		Sc.2: 5% (0.25) Vs. 5% (0.25)		Sc.3: 1% (0.25) Vs. 9% (0.25)		Sc.4: 3% (0.15) Vs. 7% (0.35)		Sc.5: 1% (0.05) Vs. 9% (0.45)	
G. cor	Method	Mean diff. (SD)	Mean B (SD)	Mean diff. (SD)	Mean B (SD)	Mean diff. (SD)	Mean B (SD)	Mean diff. (SD)	Mean B (SD)	Mean diff. (SD)	Mean B (SD)
-0.90	H-O	-0.08 (0.01)	-0.33 (0.01)	-0.43 (0.02)	-0.17 (0.01)	-0.43 (0.02)	-0.17 (0.01)	-0.62 (0.02)	-0.08 (0.02)	-0.81 (0.03)	0.01 (0.03)
	SH	-0.01 (0.02)		0.01 (0.06)		-0.12 (0.30)		-0.45 (0.51)		-0.89 (0.07)	
-0.50	H-O	-0.04 (0.01)	-0.35 (0.01)	-0.24 (0.02)	-0.26 (0.01)	-0.23 (0.02)	-0.26 (0.01)	-0.33 (0.02)	-0.22 (0.02)	-0.43 (0.03)	-0.17 (0.03)
	SH	-0.01 (0.01)		0.02 (0.08)		0.05 (0.12)		0.02 (0.07)		0.14 (0.22)	
Zero	H-O	0 (0.01)	-0.37 (0.01)	0 (0.02)	-0.37 (0.01)	0 (0.02)	-0.37 (0.01)	0 (0.02)	-0.37 (0.02)	0 (0.03)	-0.37 (0.03)
	SH	0 (0.002)		0.02 (0.02)		0 (0.01)		0.04 (0.11)		0.37 (0.12)	
0.50	H-O	0.05 (0.01)	-0.39 (0.01)	0.23 (0.02)	-0.47 (0.01)	0.23 (0.02)	-0.48 (0.01)	0.31 (0.02)	-0.51 (0.02)	0.40 (0.03)	-0.55 (0.03)
	SH	0 (0.003)		0.02 (0.07)		0 (0.04)		0.13 (0.16)		0.45 (0.17)	
0.90	H-O	0.08 (0.01)	-0.41 (0.01)	0.40 (0.02)	-0.55 (0.01)	0.40 (0.02)	-0.56 (0.01)	0.55 (0.02)	-0.62 (0.02)	0.70 (0.03)	-0.69 (0.03)
	SH	0 (0.004)		-0.03 (0.02)		0.28 (0.40)		0.72 (0.26)		0.81 (0.05)	

Abbreviations: G_1 = true class of SNPs with effects on incidence only; G_2 = true class of SNPs with effects on incidence and prognosis; G. cor = genetic correlation of SNP effects on incidence and prognosis; H-O = estimator of the method of Dudbridge et al (2019) with Hedges-Olkin adjustment for regression dilution; SH = 'Slope-Hunter' estimator; Mean diff. = mean of differences between estimated adjustment factor using corresponding estimator and the true index event bias; SD = standard deviation; B = true index event bias.

The Mean diff. represented the mean differences between adjustment factor estimated using corresponding method and the true collider bias. Under Sc.4 in this table, the Slope-Hunter method had lower differences from the true collider bias, and lower type-1 error rates, than the DHO method under all levels of genetic correlation, except under strong positive correlation.

We have revised our simulation studies and their corresponding results in this revised version. The descriptions of all simulated scenarios and their corresponding settings are reported in the Tables 1 and S1 in this revised version.

F. Finally, I wonder why the method is “sold” as an adjustment correction method instead of a causal effect estimation tool which is robust to correlated pleiotropy, which would be a far more interesting aspect. As long as an appropriate causal effect is estimated, proper trait correction can be applied as a corollary.

R[F]: Following numerous GWAS of disease susceptibility with large sample sizes, we believe that a growing number of GWAS for progression studies will be conducted, and this was our motivation in commencing this study. Our method can provide a solution for correcting the induced collider bias in such studies. This should prove helpful in paving the road for implementing case-only MR studies of a risk factor on a prognosis.

We appreciate the reviewer suggestion that our approach can be adopted in the context of MR analysis to robustly estimate the causal effect in the presence of correlated pleiotropy. This has been mentioned in the Discussion (page 16):

“The main idea of our procedure can be adopted in future in the context of the MR analysis using a large number of genetic variants including invalid instruments, particularly for experiments in which effects of instruments on exposure and outcome are correlated [26]. This potential direction may be beneficial in robustly estimating causal effects, checking violation of MR assumptions, providing probabilistic identification of the valid instruments, and detecting pleiotropy in a given problem. A few methods, e.g. the MR-mix [27] and CAUSE [28], have been recently developed with a conceptual similarity to the Slope-Hunter method in the context of MR analysis. The aim of these methods is to

use mixture models with valid and potentially correlated invalid instruments to estimate causal effect of an exposure on outcome. The Slope-Hunter approach can be adapted to identify the class of SNPs that show no pleiotropy (equivalent to class GI: in this context), and the class that demonstrates pleiotropy (class GIP in this context). This approach would likely be robust to the 'Instrument Strength Independent of Direct Effect' (InSIDE) assumption [26] but may require the ZEMPA assumption."

Reviewers' Comments:

Reviewer #1:

Remarks to the Author:

The revised draft is much better! Substantial revisions to the algorithm and the exposition have resolved most of my comments. I have only one major comment:

In Alg 1 the first step is to impose a p-value threshold to eliminate SNPs that are not associated with I. However, this threshold procedure means that eq 12 will no longer describe the distribution of the remaining SNPs -- the middle of the distribution has been removed. Why not leave all the variants in and use EM to estimate all the parameters in eq 10? It appears the only additional parameters to estimate are π_3 and π_4 . This should also eliminate the problem pointed out on the bottom of page 8 ". In the presence of these null SNPs, our model (Equation 12) is not correct, providing only an approximation of the underlying full model shown in Equation 10"

Some minor comments about the abstract:

1. I suggest a modification of the first sentence from "Studying genetic associations conditioned on a phenotype may be affected by selection bias." to "Estimates of genetic associations conditional on a heritable phenotype can be affected by collider bias". I think it is a little confusing to use selection bias here since (at least in my personal experience) that more often refers to bias induced by selecting variables using some significance criterion.

2. The sentence "Our method assumes the largest number of similar individual-SNP ratios for SNP-prognosis to SNP-incidence associations comes from the class of SNPs only affecting incidence, even if the majority of SNPs have direct effects on both incidence and prognosis." is extremely hard to understand, especially in the abstract when the reader has no introduction yet. Maybe something like "Our method assumes that a sufficient number of SNPs affect only incidence and not prognosis to form an identifiable cluster. This may be true even if the majority of SNPs have direct effects on both incidence and prognosis."

Reviewer #2:

Remarks to the Author:

I would like to congratulate the authors for the very thorough revision. They have immensely improved the m/s and the method, which is now very logical, principled and minimally ad hoc. I have only two comments left:

1. There is a typo in line 8 of the Algorithm: the last term should be " $\hat{\beta}_{gI}$ ".

2. In the real data application (BMI-adjusted insulin), both Aschard's correction (<https://www.ncbi.nlm.nih.gov/pmc/articles/PMC4320269/>) and mtCOJO could be applied (because in this case they can be applied, this is not a case-only study). It would be fair to show what results those correction would yield.

Point-by-point response

NCOMMS-20-04869B - Slope-Hunter: A robust method for collider bias correction in conditional genome-wide association studies

We are grateful to the reviewers, the senior editor (SE) and to the editor for the detailed and helpful comments and suggestions. In the following, our responses are in blue letters and the comments of the reviewers in black italic letters.

Reviewer 1 (Remarks to the Author):

The revised draft is much better! Substantial revisions to the algorithm and the exposition have resolved most of my comments. I have only one major comment:

In Alg 1 the first step is to impose a p-value threshold to eliminate SNPs that are not associated with I. However, this threshold procedure means that eq 12 will no longer describe the distribution of the remaining SNPs - the middle of the distribution has been removed. Why not leave all the variants in and use EM to estimate all the parameters in eq 10? It appears the only additional parameters to estimate are π_3 and π_4 . This should also eliminate the problem pointed out on the bottom of page 8 ". In the presence of these null SNPs, our model (Equation 12) is not correct, providing only an approximation of the underlying full model shown in Equation 10".

R: Although after p-value thresholding the estimates of genetic associations may not follow a mixture of exact normal distributions, we have showed that our algorithm delivers a convenient approximation for the mixture of Gaussian distributions, which is much easier to estimate than a mixture of truncated normal distributions. The plausibility of the approximation performed by our algorithm has been confirmed by the results. The idea of considering the four-clusters model will complicate the estimation procedure, not only because of the inclusion of additional parameters, but also because the estimation will partially rely on how successful the model is in identification of unnecessarily considered clusters ($G_{\{.P\}}$ and $G_{\{..}\}$) that do not affect incidence, and hence do not suffer from the collider bias. Throughout the development of our method, we have examined considering all the four clusters to fit the mixture model, but the maximum likelihood estimate from the EM algorithm yielded poor results.

Some minor comments about the abstract:

1. I suggest a modification of the first sentence from "Studying genetic associations conditioned on a phenotype may be affected by selection bias." to "Estimates of genetic associations conditional on a heritable phenotype can be affected by collider bias". I think it is a little confusing to use selection bias here since (at least in my personal experience) that more often refers to bias induced by selecting variables using some significance criterion.

R1: As suggested by the reviewer, we have rephrased the first sentence of the abstract to mention the collider bias. The first sentence now reads as follows "... may be affected by collider bias."

2. The sentence "Our method assumes the largest number of similar individual-SNP ratios for SNP-prognosis to SNP-incidence associations comes from the class of SNPs only affecting incidence, even if the majority of SNPs have direct effects on both incidence and prognosis." is extremely hard to understand, especially in the abstract when the reader has no introduction yet. Maybe something like "Our method assumes that a sufficient number of SNPs affect only incidence and not prognosis to form

an identifiable cluster. This may be true even if the majority of SNPs have direct effects on both incidence and prognosis.”

R2: As suggested by the reviewer and to comply with limit of words in the abstract, we have rephrased the main assumption of our method to read “... We propose a method, ‘Slope-Hunter’, that uses model-based clustering to identify and utilise the class of variants only affecting the phenotype to estimate the adjustment factor, assuming this class explains more variation in the phenotype than any other variant classes”

Reviewer 2 (Remarks to the Author):

I would like to congratulate the authors for the very thorough revision. They have immensely improved the m/s and the method, which is now very logical, principled and minimally ad hoc. I have only two comments left:

1. There is a typo in line 8 of the Algorithm: the last term should be “beta-hat_{g|}”.

R1: We have corrected this typo.

2. In the real data application (BMI-adjusted insulin), both Aschard’s correction (<https://www.ncbi.nlm.nih.gov/pmc/articles/PMC4320269/>) and mtCOJO could be applied (because in this case they can be applied, this is not a case-only study). It would be fair to show what results those correction would yield.

R2: Applying the mtCOJO method for this conditional analysis requires the marginal summary-level GWAS data for fasting blood insulin (FI), i.e. FI GWAS that is not adjusted for BMI, which is not available from the studies that we have considered in our analyses. But we have implemented the Generalised Summary-data-based Mendelian Randomisation (GSMR) method, the core phase of the mtCOJO method, which is a robust method for estimating the causal effect of a risk factor on an outcome using the summary-level GWAS data. We have estimated the collider bias by applying the GSMR method to our conditional analysis and showed the adjusted estimates in Table S11 in the supplementary material. We have added the following text to the “Results” section:

“... Applying the mtCOJO method to adjust for the collider bias in this conditional analysis requires the marginal summary-level GWAS data for FI, i.e. FI GWAS that is not adjusted for BMI, which is not available from the considered studies. We have compared the adjustments obtained by the Slope-Hunter and the DHO methods with the adjusted estimates obtained using the Generalised Summary-data-based Mendelian Randomisation (GSMR) method, the core procedure of the mtCOJO method [15], which could be used to estimate the collider bias and, through that, to derive an adjustment factor for the collider bias correction using the considered summary-level data. This enabled comparison of the adjustments of Slope-Hunter with the alternatives for exactly the same variants considered in our analyses, see supplementary Table 11.”